# Actin dynamics switches two distinct modes of endosomal fusion in yolk sac visceral endoderm cells

**Seiichi Koike[1,2,3]\*, Masashi Tachikawa[4], Motosuke Tsutsumi[5,6], Takuya Okada[1,2], Tomomi Nemoto[5,6], Kazuko Keino-Masu[1,2], Masayuki Masu[1,2]\***

[1]Graduate School of Comprehensive Human Sciences, University of Tsukuba, Tsukuba, Japan; [2]Department of Molecular Neurobiology, Institute of Medicine, University of Tsukuba, Tsukuba, Japan; [3]Laboratory of Molecular and Cellular Biology, Graduate School of Science and Engineering for Research, University of Toyama, Toyama, Japan; [4]Graduate School of Nanobioscience, Yokohama City University, Yokohama, Japan; [5]Exploratory Research Center on Life and Living Systems (ExCELLS), National Institutes of Natural Sciences, Okazaki, Japan; [6]National Institute for Physiological Sciences, National Institutes of Natural Sciences, Okazaki, Japan

## eLife assessment

This study provides **valuable** insights into the role of actin dynamics in regulating the transition of fusion models during homotypic fusion between late endosomes. The evidence supporting the authors' claims is **convincing**. However, while the observations are significant, the study could benefit from further exploration of the mechanistic details and physiological relevance.

**\*For correspondence:**
skoike@eng.u-toyama.ac.jp (SK);
mmasu@md.tsukuba.ac.jp (MM)

**Abstract** Membranes undergo various patterns of deformation during vesicle fusion, but how this membrane deformation is regulated and contributes to fusion remains unknown. In this study, we developed a new method of observing the fusion of individual late endosomes and lysosomes by using mouse yolk sac visceral endoderm cells that have huge endocytic vesicles. We found that there were two distinct fusion modes that were differently regulated. In homotypic fusion, two late endosomes fused quickly, whereas in heterotypic fusion they fused to lysosomes slowly. Mathematical modeling showed that vesicle size is a critical determinant of these fusion types and that membrane fluctuation forces can overcome the vesicle size effects. We found that actin filaments were bound to late endosomes and forces derived from dynamic actin remodeling were necessary for quick fusion during homotypic fusion. Furthermore, cofilin played a role in endocytic fusion by regulating actin turnover. These data suggest that actin promotes vesicle fusion for efficient membrane trafficking in visceral endoderm cells.

## Introduction

Eukaryotic cells contain numerous organelles that are separated from the cytoplasm by a lipid bilayer. Exchange of proteins, lipids, and metabolites between the organelles occurs through vesicle budding and fusion. To date, theoretical and computer biophysics have modeled several steps in the process of membrane fusion (*Chernomordik and Kozlov, 2008*; *Lu and Guo, 2019*). The priming of membranes is followed by the merger of the outer leaflet of the membranes to establish a hemifusion stalk. The stalk then expands into a hemifusion diaphragm and the merger of the inner leaflets leads to the

formation of a fusion pore, which expands and allows content mixing. Recently, the hemifusion stage was observed in live cells, providing strong evidence of the hemifusion model (*Zhao et al., 2016*).

Studies utilizing time-lapse microscopic observation of live cells, however, have revealed that membrane behavior after pore formation is unexpectedly divergent. For example, in giant endosomal vesicles in cells overexpressing constitutively active Rab5 protein, two different types of membrane fusion were observed (*Roberts et al., 1999*). In 'explosive' fusion, which is commonly seen in cells, the fusion pore expanded quickly. In contrast, in 'bridge' fusion, the pore did not expand; instead, one vesicle gradually shrank and disappeared over time as the result of the transfer of its vesicle content into another vesicle. In homotypic vacuole fusion in *Saccharomyces cerevisiae,* vacuole membranes fused at the vertices instead of a central pore expanding radially, and then a boundary membrane between the two docked vacuoles was released into the vacuole lumen (*Wang et al., 2002*). Because pore dynamics determine the rate of content mixing, they have significant effects on the trafficking efficiency of cargo molecules. However, how different modes of membrane deformation are induced and whether the distinct fusion modes are exchangeable remain unknown.

Membrane deformation is generally controlled by membrane-associating proteins as well as by several physical parameters including the vesicle volume, membrane area, bending energy, and osmotic energy in cells (*Farsad and De Camilli, 2003*; *Lipowsky, 1991*). Actin and its regulating proteins, via interaction with the membrane, exert physical forces on and cause fluctuation in the membrane, which eventually leads to membrane deformation by modification of the physical parameters (*Salbreux et al., 2012*). Actin produces forces in two different manners. First, assembly of branched actin filaments, nucleated from the membrane surface by actin nucleators such as the actin-related protein 2/3 (Arp2/3) complex and formins, provides pushing forces on the membrane (*Goley and Welch, 2006*; *Svitkina, 2018*). Second, actin generates pulling forces through myosin-driven sliding of actin filaments (*Svitkina, 2018*). Both forces play roles in induction of membrane deformation during vesicle fusion (*Ebrahim et al., 2019*; *Ma et al., 2020*; *Tran et al., 2015*; *Uenishi et al., 2013*). Many experiments have been carried out to elucidate the roles of actin in vesicle fusion through manipulation of actin dynamics by use of specific drugs or gene modifications. These experiments gave rise to different results depending on the experimental conditions. For example, polymerization or stabilization of actin enhanced membrane fusion in some experiments but inhibited it in others (reviewed in *Eitzen, 2003*). These different results imply that the regulation mechanisms of vesicle fusion by actin filaments are more complex than previously assumed. Moreover, how the actin forces contribute to the fusion modes remains unknown, as does how the force generated by actin polymerization or by actin-dependent motor proteins contributes to membrane deformation during fusion.

In this study, we utilized mouse yolk sac visceral endoderm (VE) cells as a model system for addressing the above questions. VE cells play a critical role in the maternofetal exchange of nutrients before the establishment of a chorioallantoic placenta: they vigorously endocytose maternal proteins, hydrolyze the proteins in lysosomes, and supply the resultant products to the developing embryo (*Bielinska et al., 1999*; *Jollie, 1990*). Reflecting this high endocytic activity, VE cells are characterized by huge endosomal vesicles (*Bielinska et al., 1999*; *Kawamura et al., 2012*; *Koike et al., 2009*). These properties of VE cells enable us to observe endosomal dynamics including vesicle fusion. Previously, by using a whole embryo culture system combined with specific pharmacologic inhibitors for intracellular signaling molecules, we showed that actin dynamics regulated by lysophosphatidic acid (LPA) receptors and the Rho-ROCK-LIM kinase pathway are required for the formation of huge lysosomes (*Koike et al., 2009*).

Here, we show that late endosomes of VE cells exhibit different modes of fusion with target organelles depending on their targets: they undergo homotypic fusion with late endosomes in an explosive fusion-like manner and heterotypic fusion with lysosomes in a bridge fusion-like manner. Our mathematical modeling predicts that the vesicle size is a critical determinant of the fusion types and that membrane perturbation overcomes the size effects. We provide evidence that actin filaments are associated with late endosomes and that actin dynamics generate forces promoting vesicle fusion of endosomes for efficient membrane trafficking.

## Results

### Endocytic vesicles in mouse yolk sac VE cells

To visualize endosomal vesicles in yolk sac VE cells, an embryonic day 8.5 (E8.5) whole embryo was labeled with a fluorescent membrane probe, FM1-43, and subsequently stained with organelle markers. The embryo was put on a glass-bottom dish, and VE cells were observed by use of confocal microscopy (*Figure 1A*). These observations revealed that different types of endosomal vesicles were aligned from the apical to the basal area in the supranuclear portion of VE cells, reflecting the repeated fusion and enlargement of vesicles as they migrated deeper into the cell (*Figure 1B and C*). In the most superficial portion (within 0.5–1 μm of the apical surface; designated as zone 1, *Figure 1—figure supplement 1A*), fuzzy FM1-43 staining was observed (*Figure 1B*). The small vesicles in this zone were positive for EEA1 (*Figure 1B*), indicating that they were early endosomes. Immediately beneath zone 1, many medium-sized round vesicles were observed (zone 2; at a depth of 1–6 μm from the apical surface, *Figure 1—figure supplement 1A*). Electroporation of EGFP-Rab7 revealed that these vesicles were positive for Rab7, suggesting that they were late endosomes. In the deep portion (zone 3; at a depth of 6–16 μm from the apical surface, *Figure 1—figure supplement 1A*), large amorphous lysosomes, positive for LysoTracker Red and LAMP1, were observed (*Figure 1B*; *Koike et al., 2009*).

To examine whether these vesicles constitute endocytic pathways, we traced vesicle trafficking in VE cells. After E8.5 whole embryos were pulse-labeled with Alexa 488-transferrin for 5 min, fluorescently labeled endocytic vesicles were tracked over time. The fluorescent signals were initially detected in small vesicles (area <2.0 μm$^2$) in zone 1; 10–20 min after labeling, the signals were mainly detected in medium-sized vesicles (2.0–25 μm$^2$) in zone 2; 40 min after labeling, the signals were strongly observed in large lysosomes (>25 μm$^2$) in zone 3 (*Figure 1D and E*). The time course of the numbers of labeled vesicles in each zone showed that endocytosed transferrin was rapidly transported from early endosomes to lysosomes via late endosomes in VE cells, as reported previously (*Richardson et al., 2000*). In addition, rhodamine-labeled dextran, which was transported by pinocytosis pathways, was also transported to lysosomes in a similar fashion (*Figure 1—figure supplement 1B*).

### Two distinct modes of late endosomal fusion

To examine the dynamic changes of endocytic vesicles, we performed time-lapse imaging of Alexa 488-transferrin- or FM 1–43-labeled vesicles. We focused on late endosomes in zone 2 because these vesicles were large enough and suitable for microscopic observation of membrane fusion processes, as described below.

First, from 5 min after pulse labeling, fusion of two neighboring late endosomes was observed frequently (*Figure 2A and B*; *Video 1*). The fusion showed an explosive mode, in which a membrane pore quickly expanded and two vesicles fused to become a single vesicle (*Figure 2B*). The size distribution of the late endosomes that underwent fusion (7.3±6.6 μm$^2$, n=34) was similar to that of the total late endosomes in the observed area at the start of recording (5.7±6.4 μm$^2$, n=265; *Figure 2C*), indicating that late endosomes of any size underwent homotypic fusion. The fusion was completed within 20–50 s (the time required from membrane fusion to single spherical vesicle formation was 31.9±0.2 s, n=37), and the time was weakly correlated with the vesicle size (*r*=0.60, *Figure 2D*).

Second, from 15 min after pulse labeling, some late endosomes shrank gradually and disappeared from the focal plane (*Figure 3A* and *Video 2*). Because a previous paper reported that late endosomes were engulfed by lysosomes in E6.5 mouse VE cells (*Kawamura et al., 2012*), we examined the mechanism of the disappearance of late endosomes in more detail using different approaches. First, we used differential labeling of late endosomes and lysosomes. When whole embryos were incubated with rhodamine-dextran for 20 min and subsequently pulse-labeled with Alexa 488-transferrin for 5 min, lysosomes and late endosomes were labeled with dextran and transferrin, respectively, at the time of observation (15 min after transferrin labeling). In a diminishing late endosome, the fluorescence intensity of transferrin in the vesicle increased initially and then decreased quickly (*Figure 3B–D*). At the same time, the rhodamine-dextran in the adjacent lysosome entered the late endosome (*Figure 3B–D*), indicating that the vesicular contents of the late endosomes and lysosomes were mixed. Second, using electron microscopy, we observed pore formation between a late endosome and a lysosome (*Figure 3E*). These data suggest that the disappearance of late endosomes is a consequence of fusion with the underlying lysosomes, but not of engulfment in E8.5 embryos. This type of fusion, so-called bridge fusion, described in a previous study (*Nicot et al., 2006*; *Roberts et al., 1999*), occurred as

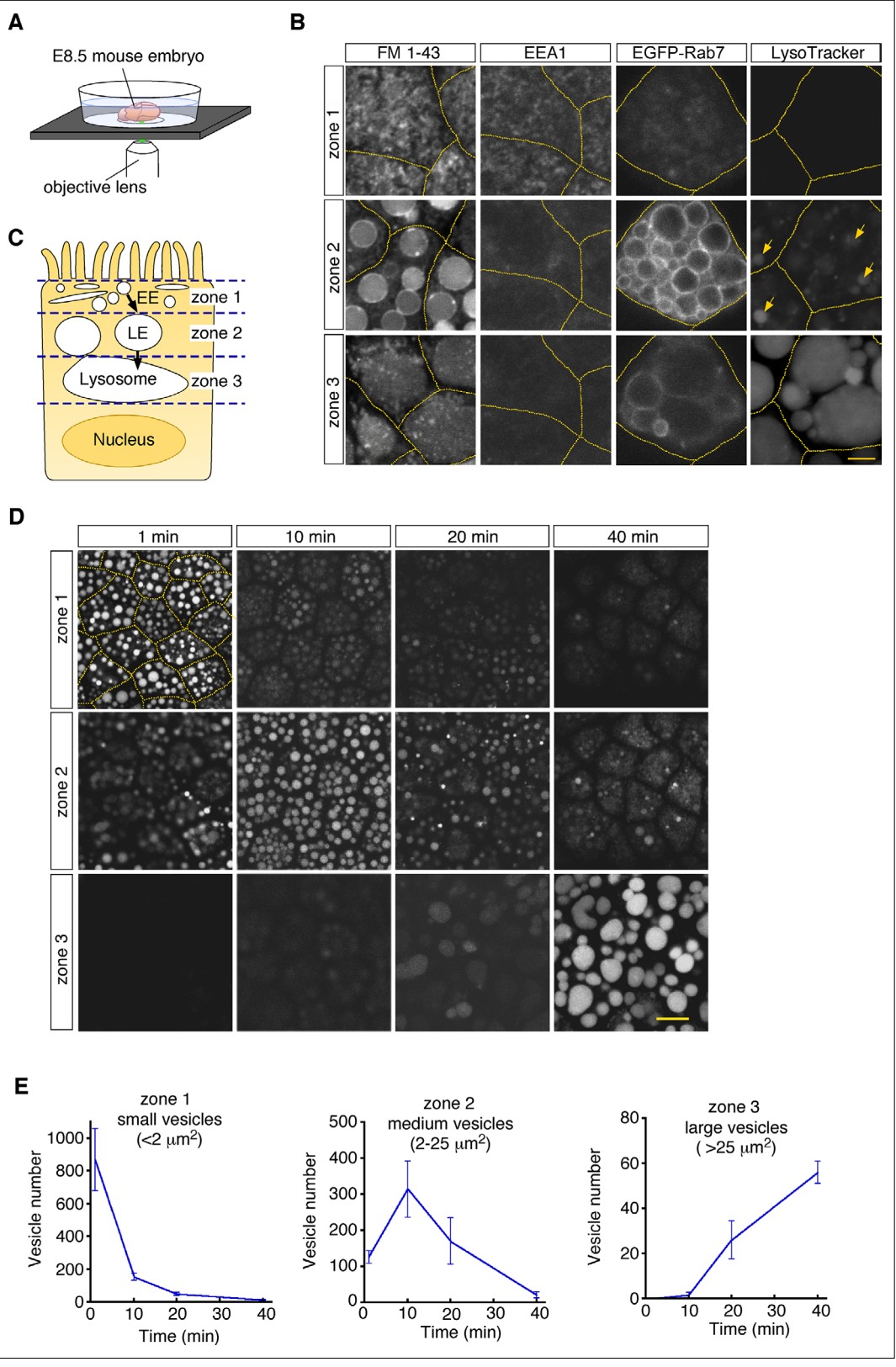

**Figure 1.** Endocytic vesicles in VE cells. (**A**) Schematic diagram of the imaging setup. An E8.5 whole mouse embryo was observed with a laser confocal microscope. (**B**) Confocal images of VE cells stained with FM 1–43, LysoTracker Red, or anti-EEA1 antibody, or electroporated with EGFP-Rab7. The yellow dotted lines indicate the cell boundaries. The yellow arrows show some vesicles that were positive for LysoTracker Red in zone 2. (**C**) For

*Figure 1 continued on next page*

*Figure 1 continued*

descriptive purposes, the supranuclear portion of VE cells is divided into three zones. Zone 1, zone 2, and zone 3 contain early endosomes (EE), late endosomes (LE), and lysosomes, respectively. (**D**) Confocal images of VE cells labeled with Alexa 488-transferrin. Images taken 1, 10, 20, and 40 min after pulse labeling are shown. The yellow dotted lines indicate the cell boundaries. (**E**) Time-course analysis of the Alexa 488-transferrin-containing vesicles in VE cells. n=4. The scale bars indicate 4 µm (**B**) and 10 µm (**D**).

The online version of this article includes the following figure supplement(s) for figure 1:

**Figure supplement 1.** Vesicular transport of dextran to lysosomes.

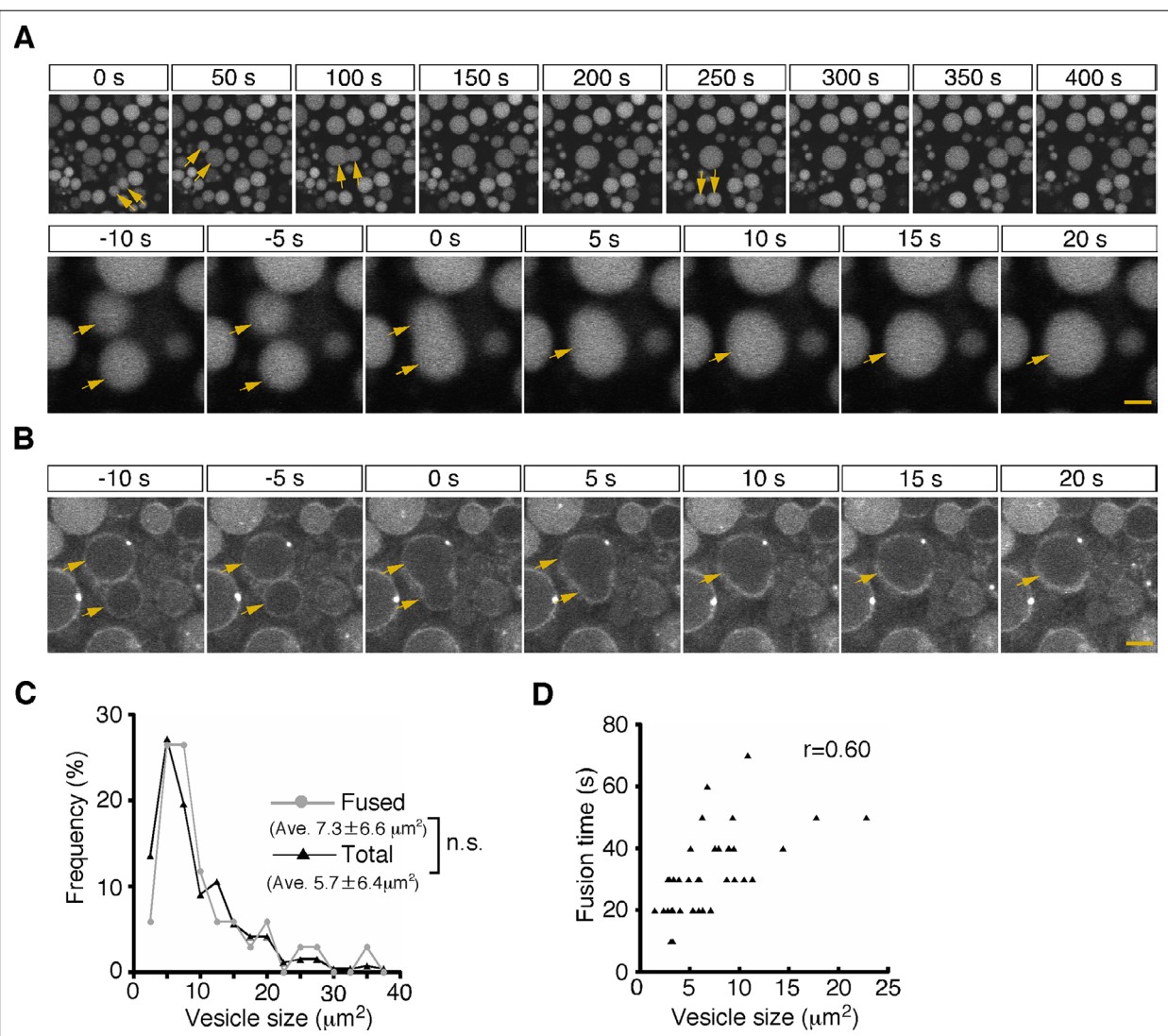

**Figure 2.** Homotypic fusion between late endosomes. (**A**) Time-lapse imaging of endocytic vesicles in VE cells. After 5 min of pulse-labeling with Alexa 488-transferrin, homotypic fusion of late endosomes (arrows) was frequently observed in zone 2. In the upper panel, time 0 indicates the start of time-lapse imaging. In the lower panel, time 0 refers to the start of the fusion of the endosomes indicated by arrows. (**B**) By labeling of VE cells with FM1-43, the fusion process of the cell membranes in zone 2 was observed. Time 0 refers to the start of the fusion of the endosomes, indicated by arrows. (**C**) Histograms showing the size distribution of total late endosomes (black line, n=265) and the late endosomes that underwent homotypic fusion (gray line, n=37). No significant difference was observed between the two groups (Mann-Whitney U test). (**D**) Correlation between the size of the late endosomes that underwent homotypic fusion and the time required for completion of fusion (i.e. the time required from membrane fusion to formation of a single round vesicle). The sizes of the larger of the fused vesicles are plotted. Thirty-seven fusion events were measured. The scale bars indicate 5 µm (A, top) and 1.5 µm (A, bottom; **B**).

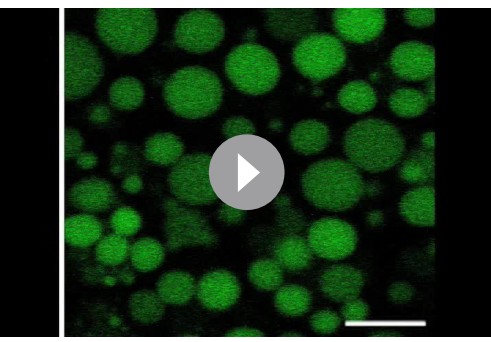

**Video 1.** Homotypic fusion of late endosomes in yolk sac VE cells. Time-lapse recording of homotypic fusion of late endosomes. At 5 min after labeling with Alexa Fluor 488-transferrin, homotypic fusion of late endosomes was observed frequently in the VE cells. The scale bar indicates 10 μm.

https://elifesciences.org/articles/95999/figures#video1

a result of the slow mixing of vesicular contents through a narrow fusion pore. Actually, heterotypic fusion was slower than homotypic fusion and was completed within 90–250 s (154.8±1.4 s, n=25), and the fusion time (the time required from membrane fusion to disappearance of the vesicle) did not correlate with the vesicle size (*Figure 3F*). The size of the labeled late endosomes at this time point (10.1±8.1 μm², n=256) was larger than that at 5 min after labeling (6.7±6.2 μm², n=242), indicating that larger late endosomes tended to undergo heterotypic fusion (*Figure 3G*, see also below). However, the size of the late endosomes that underwent heterotypic fusion (10.2±9.2 μm², n=34) did not differ from that of the total late endosomes at 15 min after labeling (*Figure 3G*), indicating that the size of late endosomes is not the sole trigger for heterotypic fusion.

## Vesicle size is an important factor for the transition of fusion modes

To examine whether the differences in membrane deformation of late endosomes depend on their fusion targets, we adopted a mathematical modeling approach that describes the membrane deformation processes after pore formation. According to thermodynamics, the deformation of a physical object proceeds as its free energy decreases. The free energy of a vesicle consists of two terms: bending energy and osmotic energy. The bending energy, $b$, is the integral of the square of the mean curvature all over the surface (*Helfrich, 1973*). The osmotic energy is the product of the osmotic pressure difference between the inside and the outside of the vesicle, $\Delta p$, and the vesicle volume, $V$. Thus, the free energy is given by

$$F = b - \Delta p \cdot V.$$

Note that the bending energy depends only on its shape; in other words, two membranes with the same shape but with different sizes have the same bending energy. On the other hand, the volume depends on both the size and the shape. Therefore, the vesicle size intrinsically affects the balance of both energy sources and may be a critical factor for membrane deformation during fusion.

We focused on a closed membrane having a constricted neck region and introduced two shape parameters, the neck width, $w$, and the area ratio between both sides of the neck, $a$ (*Figure 4A*), so that the basic processes of the two fusion types are described by the changes in $w$ and $a$ just after pore formation, $w = 0$. The relaxation of the neck in explosive fusion is expressed by the increase in $w$. On the other hand, the absorption of smaller vesicles by the larger one in bridge fusion is the decrease in $a$ while $w$ remains small.

When calculating the free energy as a function of $w$ and $a$, the deformation is expressed as the process of descending the free-energy landscape in the space. Because the shape giving the minimum free energy is most likely to be realized, we first specified the minimum-free-energy shape and then calculated the free energy value for each parameter set.

We considered two extreme conditions: the membrane size is sufficiently small or large. In the former case, the osmotic energy is negligible and the free energy is approximated to

$$F \simeq F^{small} = b$$

The free-energy landscape on the shape parameter space $(w, a)$ is shown in *Figure 4B*. In the latter case, although the osmotic energy is dominant, the bending energy is also not negligible because the bending energy at the neck region becomes considerably large. Thus, the free energy for the large membrane case is

$$F \simeq F^{large} = B^{neck} - \Delta p \cdot V,$$

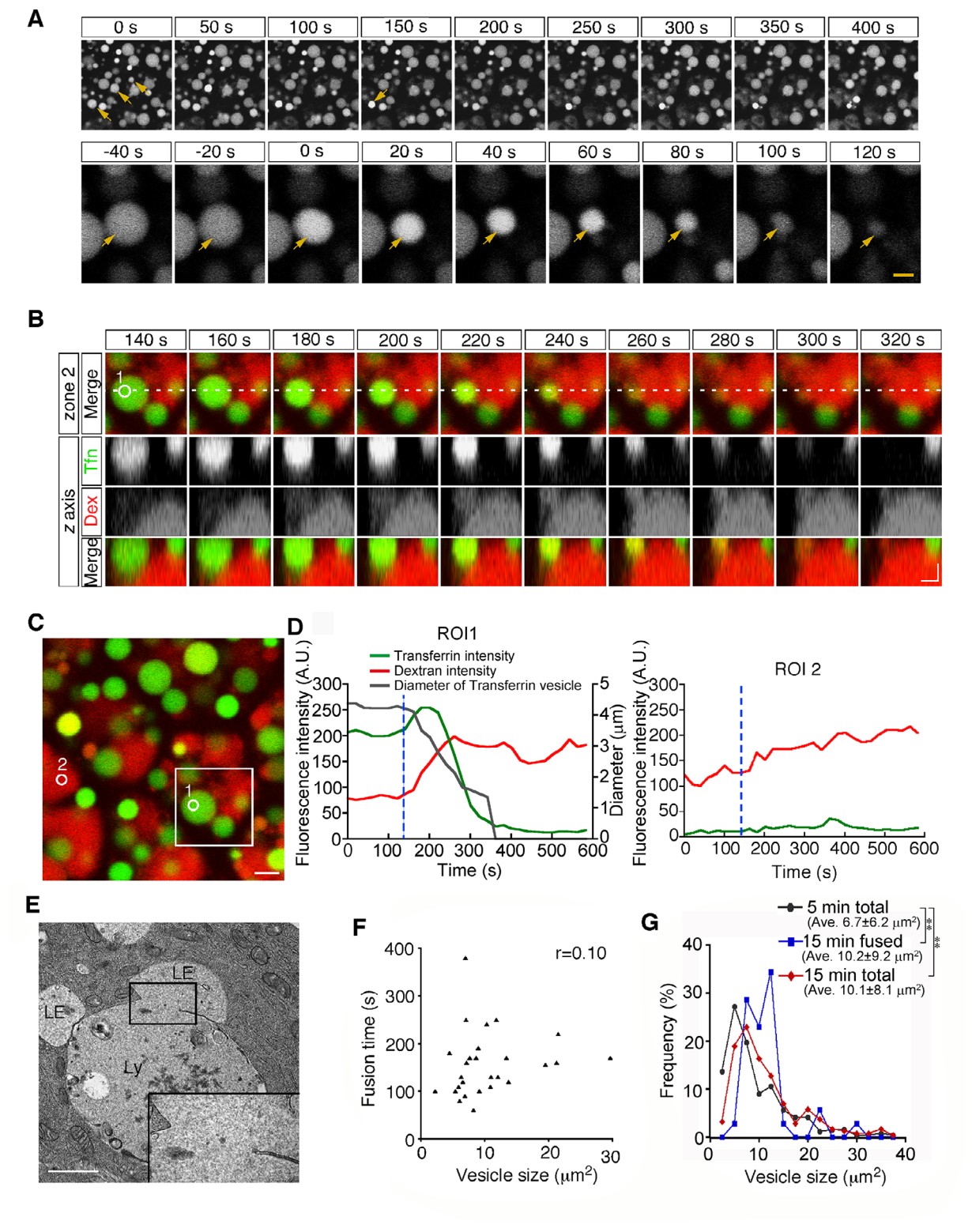

**Figure 3.** Heterotypic fusion between late endosomes and lysosomes. (**A**) Time-lapse imaging of VE cells. After 15 min of pulse-labeling with Alexa 488-transferrin, heterotypic fusion of late endosomes was observed: they shrank gradually and disappeared from the focal plane (arrows). In the upper panel, time 0 indicates the start of time-lapse imaging. In the lower panel, time 0 refers to the start of the fusion of the endosomes indicated by an arrow. (**B**) Time-lapse imaging of late endosomes in VE cells. Late endosomes and lysosomes were labeled with Alexa 488-transferrin (Tfn, green) and rhodamine-dextran (Dex, red), respectively. The bottom figures show optical sectioning through the dotted line in the upper figure. Time 0 indicates

*Figure 3 continued on next page*

*Figure 3 continued*

the start of time-lapse imaging. (**C, D**) Time course of the fluorescence intensity. In a late endosome undergoing fusion (ROI 1 in [B-C]), the fluorescence intensity of transferrin (green) slightly increased and then quickly decreased after membrane fusion. The fluorescence signal of dextran (red) in an underlying lysosome leaked into the fusing late endosome. The white box in (**C**) indicates the area shown in (**B**). The gray line in (**D**) indicates the diameter of the transferrin-positive late endosomes. The blue dotted lines in (**D**) indicate the moment the fusion pore was formed in ROI 1. ROI 2 is a negative control in a lysosome that did not undergo fusion. (**E**) Electron microscopic image showing pore formation between a late endosome (LE) and a lysosome (Ly). The inset shows a magnified picture of the boxed region. (**F**) Correlation between the size of late endosomes that underwent heterotypic fusion and the time required for completion of fusion (i.e. the time from the start of shrinkage to disappearance). Twenty-five fusion events were measured. (**G**) Histograms showing the size distribution of late endosomes at 5 min (gray line, n=256) and 15 min (red line, n=242) after labeling with Alexa 488-transferrin, as well as the late endosomes that underwent heterotypic fusion at 15 min (blue line, n=34). The average size of the vesicles was larger at 15 min than at 5 min (\*\*p<0.01, Mann-Whitney U test). However, the size distribution of the late endosomes that underwent heterotypic fusion at 15 min did not differ from that of the total late endosomes at 15 min. The scale bars indicate 5 µm (A, top), 1.5 µm (A, bottom), 2 µm (**B**), 3 µm (**C**), and 1 µm (**E**).

where $B^{neck}$ is the bending energy at the neck. The free-energy landscape for a vesicle with the projected area of $10 \left[ \mu m^2 \right]$ is shown in *Figure 4C*.

These free-energy landscapes (*Figure 4B and C*) clearly differ from each other, implying that they drive different deformation processes. In particular, the contour of the $F^{small}$ landscape around $w$ 0 is perpendicular to the $w$ axis, whereas the contour of the $F^{large}$ landscape is parallel to the $w$ axis. Typical orbits of the shape deformations from the initial condition $(w, a) = (0.0, 0.3)$ generated by the gradients of the landscape are shown as gray arrows in *Figure 4B and C*. The orbit in the $F^{small}$ landscape is characterized by a smooth increase in $w$; that is the neck expands quickly (*Figure 4D*), resulting in explosive fusion. On the other hand, in the $F^{large}$ landscape, $a$ increases through the orbit, whilst $w$ remains small, which evokes bridge fusion (*Figure 4E*).

The deformation processes observed in our modeling seem to be qualitatively consistent with our experimental results showing that the size of the fusion target changes the fusion modes (*Figures 2 and 3*). Next, we made a rough estimate of how much of the vesicle size the mathematical model expects to switch the types of fusion. The typical bending energy of a (spherical) vesicle is $b \simeq 2 \times 10^{-18} \left[ J \right]$ (*Evans and Rawicz, 1990*). Referring to the density differences in the dominant ions of endosomes and cytoplasm (*Scott and Gruenberg, 2011*), we assume that the total ion difference is in the range of $1 \sim 10 mM$, and that the osmotic pressure difference is $\Delta p \simeq 2.5 \sim 25 \left[ Pa \right]$. From these assumptions, $0.22 \sim 1.0 \left[ \mu m^2 \right]$ would be the vesicle size to switch fusion modes. However, the size of the observed late endosomes (5–20 µm² in *Figure 3F*) is much larger than the predicted size, suggesting that homotypic fusion between late endosomes should show the bridge fusion mode.

To resolve these inconsistencies, we considered the fluctuation effects. Intracellular membranes receive various fluctuations including thermal noise and active fluctuation from the cytoskeletons and motor proteins (*Biswas et al., 2017*; *Chen et al., 2009*). Although orbits descending the $F^{large}$ landscape are energetically restricted to keep $w$ small, these fluctuations may force the orbits to deviate from the restricted path (the dashed arrow in *Figure 4C*); that is they are expected to induce explosive fusion. Then, we performed Monte-Carlo simulations to generate orbits descending the free-energy landscape with various magnitudes of fluctuations and determined the probabilities of orbits that deviate from the path (*Figure 4F*). When we consider only the thermal noise with room temperature with the estimated osmotic pressure difference ($\Delta p \simeq 2.5 \sim 25 \left[ Pa \right]$), as for late-endosome-sized vesicles ($2.0 \sim 25 \left[ \mu m^2 \right]$), the simulation predicts that vesicles will show bridge fusion (the

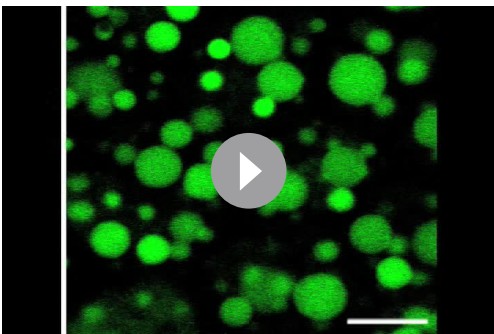

**Video 2.** Heterotypic fusion of late endosomes in yolk sac VE cells. Time-lapse recording of heterotypic fusion between late endosomes and lysosomes. At 15 min after labeling with Alexa Fluor 488-transferrin, late endosomes often shrank and disappeared from the focal plane as a result of heterotypic fusion of late endosomes with pre-existing lysosomes that were out of the focal plane. The scale bar indicates 10 µm.
https://elifesciences.org/articles/95999/figures#video2

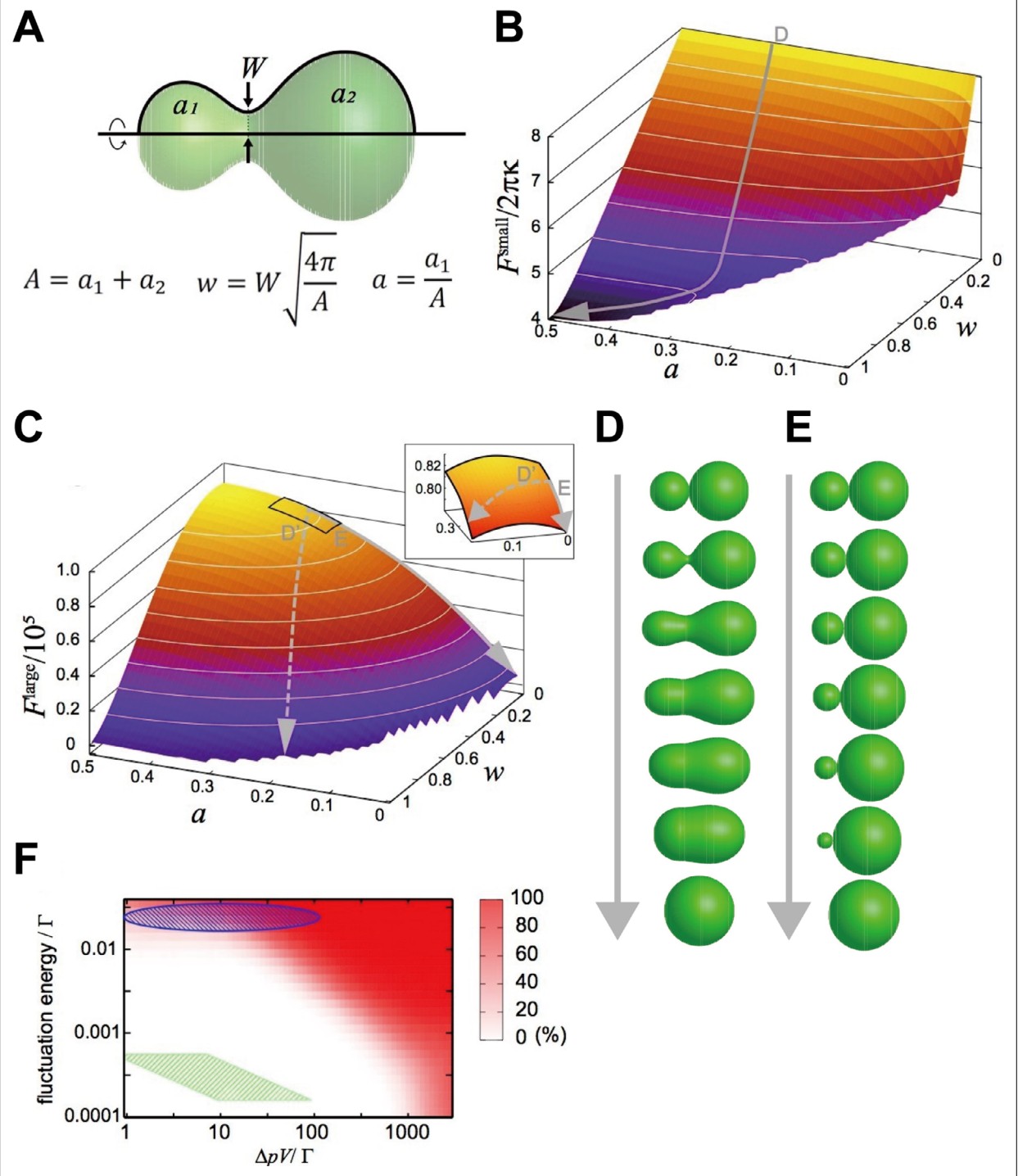

**Figure 4.** Mathematical modeling analysis of membrane deformation after connection of 2 vesicles. (A) Definition of shape parameters. (B) Free-energy landscape for small vesicles. The gray arrow (D) indicates a typical orbit of membrane deformation from the initial condition $(w, a) = (0.0, 0.3)$. (C) Free-energy landscape for large vesicles. The solid gray arrow (E) indicates the typical orbit of membrane deformation, whereas the dashed gray arrow (D') indicates the typical orbit of membrane deformation under fluctuated conditions. The inset is a blowup of the region $w = 0 \sim 0.15$, $a = 0.25 \sim 0.35$. (D, E) Time series of shape variation on the typical orbits indicated in (B) and (C), respectively. Note that the exact time scale cannot be estimated in this analysis. (F) Probability of the explosive fusions in the $F^{large}$ landscape (the orbit of the dashed gray arrow in [C]) for various vesicle sizes, osmotic pressure differences, and fluctuation energies, estimated by Monte-Carlo simulations. $\Gamma = 2\pi\kappa\sqrt{A}/R_0$ with the bending modulus $\kappa = 10 \, [k_B T]$ and the minimum bending radius $R_0 = 100 \, [nm]$. The green shaded region indicates the condition: $\Delta p \simeq 2.5 \sim 25 \, [Pa]$, $V = 4A\sqrt{A}/3\sqrt{\pi}$ with $A = 2 \sim 25 \, [\mu m^2]$ and the fluctuation energy $1 \, [k_B T]$ (thermal fluctuation at room temperature). The green shaded region corresponds to late-

*Figure 4 continued on next page*

*Figure 4 continued*

endosome-sized vesicles ($2.0 \sim 25\ [\mu m^2]$) under estimated osmotic pressure difference ($\Delta p \simeq 2.5 \sim 25\ [Pa]$) and the thermal noise at room temperature. The blue-shaded region indicates the vesicles under larger fluctuations.

The online version of this article includes the following figure supplement(s) for figure 4:

**Figure supplement 1.** Mathematical modeling of membrane deformation during membrane fusion.

green-shaded region in *Figure 4F*). However, if we take into account the fact that a larger fluctuation allows explosive fusion (the blue-shaded region in *Figure 4F*), fluctuations generated by the cytoskeleton and motor proteins, much larger than thermal noise, may induce explosive fusion.

## Actin filaments are bound to late endosomes in VE cells

Previously we found that lysosome biogenesis was regulated by actin dynamics in VE cells (*Koike et al., 2009*). To examine the relationship between actin filaments and endocytic vesicles, actin filaments were visualized by use of Alexa 488-phalloidin. Endocytic vesicles were visualized by immunostaining with anti-mouse IgG because IgG is endocytosed from the maternal tissue in utero and can be stained without incubation of embryos in vitro for the pulse labeling of endosomes. In VE cells, strong signals were detected to be associated with the cell membrane (*Figure 5A–C*). In addition, in zone 1, fuzzy cytoplasmic staining was observed (*Figure 5A*), which appeared to correspond to the actin meshwork in the terminal web. In zone 2, small punctates and short filaments were detected in the cytoplasm (*Figure 5B*) and were bound to the surface of late endosomes (*Figure 5B, B' and B"*) and infrequently to lysosomes in zone 3 (*Figure 5C, C' and C"*). Optical cross-sectional images and 3D reconstruction showed that several rod-like actin filaments, extending from the apical side of VE cells into the cytoplasm, surrounded and held the late endosomes (*Figure 5D–F*). These data indicate that most actin punctates observed in the x-y plane correspond to the cross-sectional images of actin filaments extending in the apical-to-basal direction. In addition, when a fusion protein of an actin-bundling protein, fascin-1, with a red fluorescent protein was expressed in VE cells by electroporation, fascin-1 was highly colocalized with the actin punctates, suggesting that most of the actin punctates surrounding the late endosomes were bundled (*Figure 5G*).

To investigate the endosome-associated actin filaments in more detail, we performed high-resolution imaging analysis using a deconvolution technique, which had resolution comparable to super-resolution microscopy. It demonstrated that fine actin filaments extended radially from the endosomal membranes in the x-y plane, in addition to the actin filaments extending in the apical-basal axis (*Figure 5H*).

To examine the dynamics of actin filaments associated with late endosomes, E7.5 whole embryos were electroporated with a pEGFP-actin vector, and the EGFP signals in VE cells were observed by use of confocal microscopy 24 hr after electroporation. Because high levels of actin overexpression led to the deterioration of cell functions (see white arrows in *Figure 6—figure supplement 1A*), cells expressing low levels of EGFP-actin were selected for analysis: those selected cells were normal in morphology and endocytosis (see white arrowheads in *Figure 6—figure supplement 1A*). Punctate signals of EGFP-actin were seen on the late endosomal membrane similarly to those observed in Alexa 488-phalloidin labeling. Time-lapse imaging showed that some new spots had appeared, others had disappeared, and two actin spots were unified on the membrane surface of endocytic vesicles (*Figure 6A*). To examine the turnover rate of actin spots in VE cells, we performed fluorescence recovery after a photobleaching (FRAP) assay. After EGFP-actin on a late endosome was photobleached, recovery of its fluorescence was measured (*Figure 6B*). Recovery of actin spots on late endosomes occurred more rapidly than the control recovery measured at the cell surface ($t_{1/2}$ = 29.0 s; *Figure 6C–E*). Especially, the recovery of actin spots in the apical area of late endosomes was faster than in the basal area ($t_{1/2}$ = 11.5 s in the apical area vs. $t_{1/2}$ = 18.9 s in the basal area; *Figure 6C and E*). These results indicated that actin turnover on the surface of late endosomes was dynamically regulated, with a tendency to be more active in the apical area than in the basal area. The time-lapse images showed that when late endosomes moved in an x-y plane in zone 2, spots of EGFP-actin were always at the rear-end of the endosomes along the moving direction (*Figure 6F*, n=13). Because it is known that Arp2/3-dependent actin polymerization on the membrane acts as the driving force of the motility of intracellular pathogens (*Welch and Way, 2013*) and endosomes/lysosomes in *Xenopus* egg

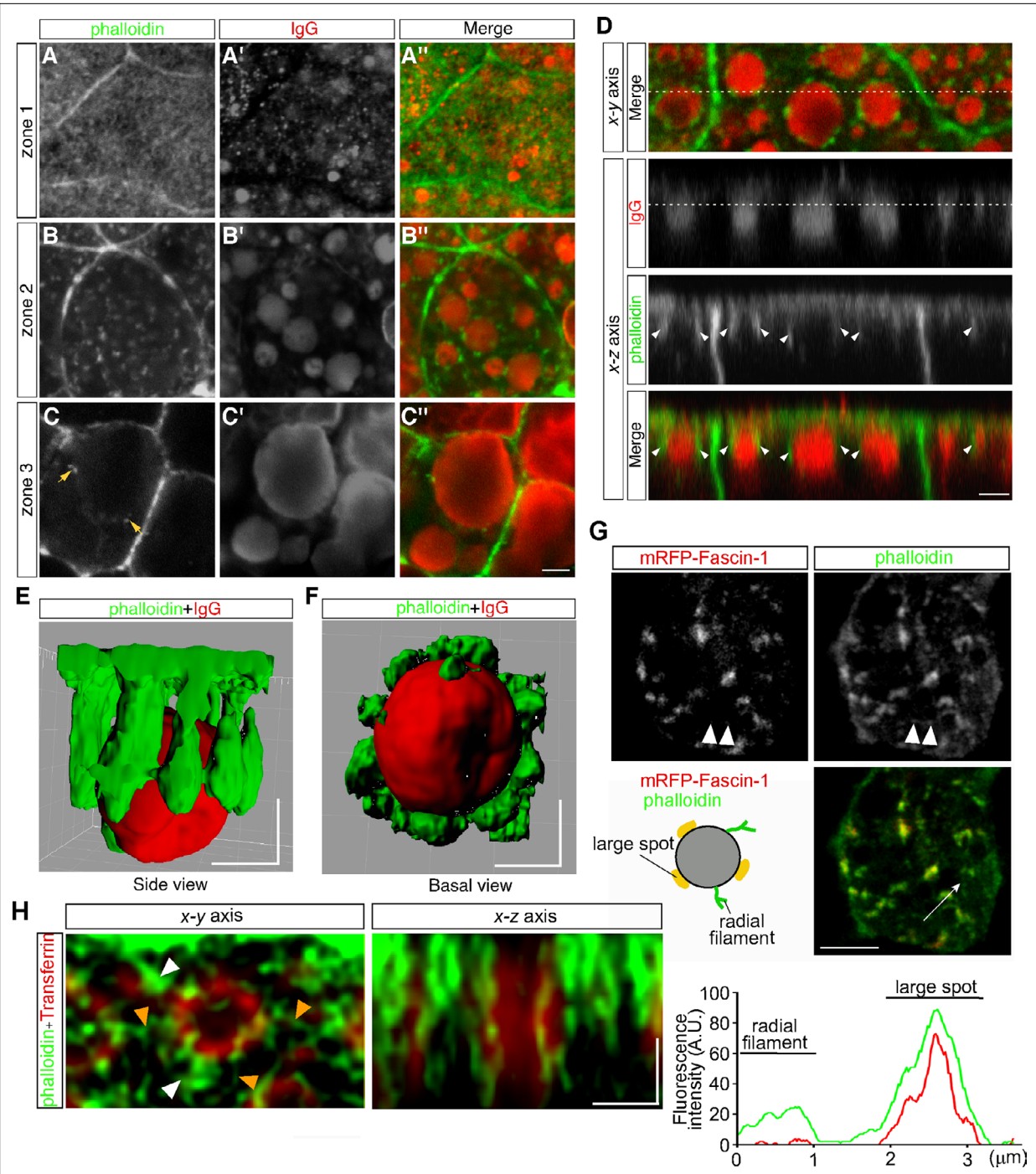

**Figure 5.** Actin filaments are associated with late endosomes in VE cells. (**A–C**) Confocal images of E8.5 mouse embryos stained with Alexa 488-phalloidin and anti-mouse IgG. Mouse IgG was incorporated into late endosomes and used as a marker for endosomes. Punctate cytoplasmic staining of phalloidin (green) was associated with the IgG-containing endocytic vesicles (red) in zone 2 (B–B") and at lower frequencies with the lysosomes in zone 3 (arrows, C–C"). (**D**) Optical sectioning of VE cells. The cross-sections along the x-z axis of the area indicated by the dotted line in the upper panel (x-y axis) are shown. The dotted line in the IgG panel represents the apical region of zone 2. Actin filaments, extending from the apical surface into the cytoplasm (white arrowheads), were closely associated with the endocytic vesicles (red). (**E, F**) Three-dimensional surface rendering of actin filaments and a late endosome. The side view (**E**) and basal view (**F**) are shown. (**G**) Alexa 488-phalloidin staining of mRFP-fascin-1-expressing VE cells. Actin punctates were highly colocalized with fascin-1, whereas fine actin filaments extending radially in the x-y plane from endosomes were negative for mRFP-fascin-1 (white arrowheads). The fluorescence intensity plot along the white arrow in the middle panel shows colocalization of fascin-1 and phalloidin in the large spot. (**H**) Deconvolution images of late endosomes and actin. Maximum intensity projection images are shown. In addition to strong phalloidin-positive spots (white arrowheads), which were observable by use of conventional confocal microscopy, actin filaments extending

*Figure 5 continued on next page*

Figure 5 continued
from the endosomal surface (orange arrowheads) were visualized in the x-y axis image. In the x-z axis image, a late endosome was surrounded by actin filaments, which extended from the apical cell surface to the basal side in the cytoplasm. The scale bars indicate 3 μm (A–D), 5 μm (**G**), and 1 μm (E–F), (**H**).

extracts (*Taunton et al., 2000*), we examined the localization of Arp2/3 in VE cells to test the possible role of Arp2/3 in endosome migration. When a fusion protein of Arp3-EGFP was expressed in VE cells by electroporation, the EGFP signals were distributed in dots on the late endosomes and partially colocalized with phalloidin spots in the x-y plane in zone 2 (*Figure 6—figure supplement 1B*). In addition, Arp3-EGFP signals were always seen at the rear-end of the endosomes along the moving direction (*Figure 6—figure supplement 1C*). Taken together, these data suggest that the radial branched filaments are temporally polymerized to provide the propulsive force on the membrane for intracellular movement.

## Actin dynamics regulate homotypic fusion of late endosomes

To examine the roles of actin in late endosomal fusion, we tested whether perturbation of actin dynamics caused defects in vesicle fusion. Treatment with cytochalasin D (a cell-permeable inhibitor of actin polymerization) for 5 min significantly changed the distribution of actin filaments. High-resolution imaging analysis using a deconvolution technique showed that phalloidin-positive filamentous structures decreased and instead actin aggregation was observed in the cytosol (*Figure 7—figure supplement 1A*). The treatment reduced the frequency of homotypic fusion in a dose-dependent manner (*Figure 7A and B*; *Video 3*). In addition, it altered the speed and mode of the fusion. Treatment with 0.1 or 1.0 μM cytochalasin D slowed down homotypic fusion (*Figure 7C and D*; *Figure 7—figure supplement 1B*). After membrane fusion, two endosomes remained in a peanut shape for a long time, indicating slower neck expansion than that in the control cells (*Figure 7—figure supplement 1B*). FM 1–43 labeling revealed that the membranes were rapidly fused, whereas it took a long time for the gourd-shaped vesicles to form a large round vesicle (*Figure 7—figure supplement 1B*). Furthermore, upon 1.0 μM cytochalasin D treatment, about one-third of homotypic fusion showed bridge fusion (*Figure 7C and E*; *Video 4*). Therefore, cytochalasin D slowed down the fusion processes and changed the fusion mode to bridge fusion at higher concentrations. These data are consistent with our mathematical model showing that larger fluctuation is necessary for explosive fusion in homotypic fusion of late endosomes (*Figure 4F*). Treatment with jasplakinolide (a cell-permeable stabilizer of actin filaments) also changed the actin dynamics. After the treatment, the actin meshwork beneath the plasma membrane (zone 1) was denser (*Figure 7—figure supplement 2*). Furthermore, actin filaments extending radially from the late endosomes were thicker and the fluorescence intensity was increased, although the actin filaments extending from the apical-basal axis were not changed (*Figure 7—figure supplement 2*). The treatment reduced the frequency of homotypic fusion in a dose-dependent manner (*Figure 7A and B*; *Video 5*), indicating that dynamic actin turnover is required for homotypic fusion.

We next observed the distribution of actin filaments during homotypic fusion. When late endosomes underwent fusion, EGFP-actin punctates were excluded from the membrane docking sites and highly polymerized transiently at both ends of the fusing vesicles (opposite to the docking region) during the rounding period after membrane fusion (*Figure 7F*). The fluorescence intensity of EGFP-actin on endosomes was quantitated in the proximal, lateral, and distal regions at the predocking, docking, rounding, and postfusion stages. This analysis indicated that actin filament distribution was spatially regulated during fusion (*Figure 7G*, dark gray bar), whereas FM 1–43 distribution, used as an unchanged control, was uniform on the membrane throughout the rounding stages (*Figure 7G*, light gray bar). Furthermore, the total fluorescence intensity of EGFP-actin on late endosomes significantly increased transiently after pore formation (*Figure 7H*), indicating that actin filaments were polymerized temporarily, not simply distributed differently. However, cytochalasin D treatment stabilized EGFP-actin spots and thereby reduced movement of EGFP-actin spots and temporal polymerization during homotypic fusion (*Figure 7—figure supplement 3*). Moreover, accumulation of Arp3-EGFP in the distal region during the rounding stage was observed similarly to EGFP-actin accumulation (*Figure 7—figure supplement 4*), and treatment with CK-666, an Arp2/3 inhibitor, resulted in a decrease in the frequency of homotypic fusion (*Figure 7B*) and slowed down homotypic fusion

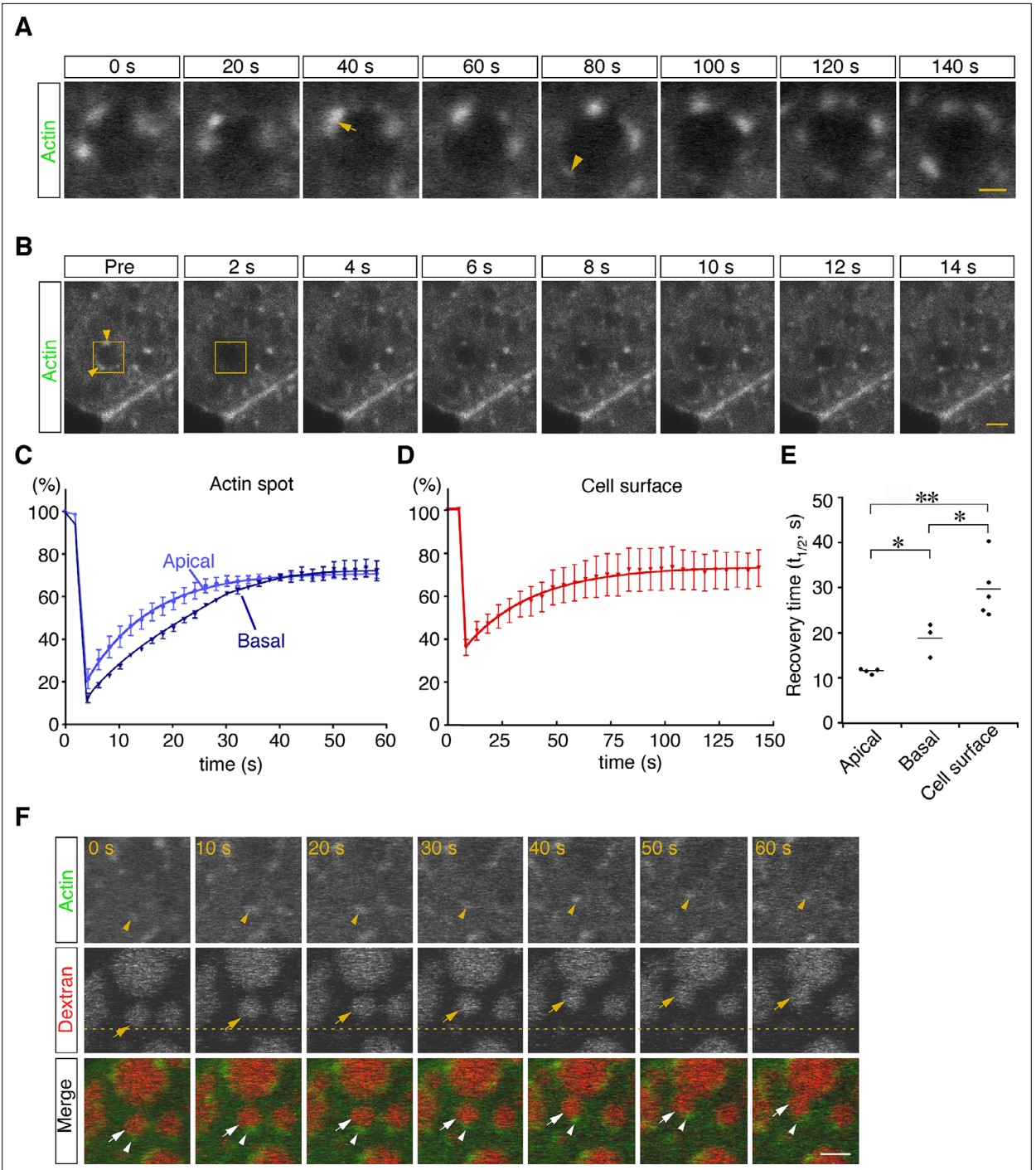

**Figure 6.** Dynamic regulation of actin filaments surrounding late endosomes. (**A**) Representative time-lapse images of EGFP-actin spots on a late endosome. VE cells were electroporated with EGFP-actin at E7.5, and time-lapse analysis was performed on the next day. The arrow indicates the fusion of actin spots. The arrowhead indicates the appearance of a new actin spot. Time 0 refers to an arbitrary time point in time-lapse imaging. (**B**) The turnover rate of actin was analyzed by use of a FRAP assay. The area (4 μm x 4 μm) covering a single late endosome at the apical region of zone 2 of VE cells expressing EGFP-actin was bleached, and the fluorescence recovery was observed by use of time-lapse microscopy. The yellow boxes indicate the laser-irradiated area, and the yellow arrowheads, actin spots recovered after photobleaching. The time laser irradiation was ended is indicated as 0. (**C, D**) Graphs showing the fluorescence recovery of EGFP-actin surrounding late endosomes at the apical region of zone 2 (light blue in [C], n=4) and the basal region of zone 2 (dark blue in [C], n=3) or at the cell surface of VE cells (D, n=5). Fluorescence intensity was normalized to prebleaching intensities. (**E**) Half-recovery time (t$_{1/2}$) of EGFP-actin fluorescence at the apical and basal regions of zone 2 and at the cell surface. Data were analyzed by use of one-way ANOVA with the Tukey multiple comparison test (*p<0.05; **p<0.01). (**F**) EGFP-actin localization during vesicle movement. Endocytic vesicles

*Figure 6 continued on next page*

*Figure 6 continued*

(red) visualized by use of rhodamine-dextran in zone 2 are shown. When a vesicle (arrow) moved in a certain direction (upward in this figure), the actin patch (arrowhead) was observed at the rear end of the movement direction. The dotted line indicates the initial position of the vesicle. Time 0 refers to an arbitrary time point in time-lapse imaging. The scale bars indicate 1 µm (**A**) and 3 µm (**B, F**).

The online version of this article includes the following figure supplement(s) for figure 6:

**Figure supplement 1.** EGFP-actin electroporation and colocalization of Arp3-EGFP with actin filaments surrounding late endosomes.

(*Figure 7C and D*). These results suggest that branched actin filaments nucleated by the Arp2/3 complex on the distal surface of late endosomes are necessary to transform the vesicle into a large round sphere quickly after membrane fusion of late endosomes.

## Homotypic and heterotypic fusions are differentially regulated by actin

We next examined the role of actin filaments in heterotypic fusion. Cytochalasin D treatment led to a significant reduction in the frequency of heterotypic fusion in a dose-dependent manner (*Figure 8A–C* and *Video 6*). In contrast, jasplakinolide treatment did not inhibit heterotypic fusion and somehow increased the frequency of heterotypic fusion with lysosomes at as early as 5 min after pulse labeling (*Figure 8B* and *Video 7*). In addition, CK-666 treatment had no effects, suggesting that Arp2/3 was not required for heterotypic fusion (*Figure 8C*). These data suggest that actin played distinct roles in homotypic fusion and heterotypic fusion and that stable actin filaments are implicated in driving heterotypic fusion.

Myosins are a superfamily of actin-based motor proteins and their members are grouped into many classes according to their structures and functions (*Hartman and Spudich, 2012*). To determine which class of myosins is important for vesicle fusion and motility, we examined the effects of inhibitors for several myosin classes that are known to be involved in vesicle transport. We tested inhibitors of myosin I (pentachloropseudilin, PCIP), nonmuscle myosin II (NM II) (blebbistatin), myosin V (MyoVin-1), and myosin VI (2,4,6-triiodophenol, TIP; *Bond et al., 2013*). Heterotypic fusion frequency was reduced to 20% by MyoVin-1 and to a lesser extent, but significantly, by blebbistatin, but was not affected by PCIP or TIP (*Figure 8D*). In contrast, homotypic fusion frequency was significantly reduced by PCIP, MyoVin-1, and TIP (*Figure 8—figure supplement 1*). EGFP-myosin IIA signals overlapped with phalloidin-positive spots on late endosomes (*Figure 8E*), whereas EGFP-myosin Va signals were juxtaposed to phalloidin-positive spots on late endosomes (*Figure 8E*).

## Cofilin-dependent actin dynamics are involved in late endosome fusion

Late endosomes undergo homotypic fusion almost exclusively at the beginning and then switch to heterotypic fusion over time after endocytosis (*Figure 9A*). Given that actin plays a different role in homotypic fusion and heterotypic fusion and that pharmacologic changes in actin dynamics shift the fusion mode as shown above, it is suggested that actin dynamics are a critical factor for the transition of fusion modes. Thus, we next decided to look into the regulation of actin dynamics in two different fusion modes. Previously, we found that lysosome biogenesis was regulated by actin dynamics through cofilin downstream of the autotaxin–LPA receptor–Rho–ROCK–LIM kinase pathway in VE cells (*Koike et al., 2009*). When VE cells were incubated with the S3 peptide, which contains the N-terminal 16 amino acid sequence of cofilin fused to a cell permeation sequence and is used as a competitive inhibitor for LIM kinase and serves as a cofilin activator (*Maekawa et al., 1999*; *Aizawa et al., 2001*), lysosomes were fragmented in VE cells and the fluorescence intensity of phalloidin in VE cells was reduced (*Koike et al., 2009*).

We thus focused on cofilin in endosomal fusion in VE cells. YFP-cofilin showed small punctate staining that colocalized with phalloidin-positive spots in the x-y plane and extended along the phalloidin signal (*Figure 9B and C*). When the fluorescence intensity of Alexa 546-phalloidin- and YFP-cofilin-positive filaments was quantitated along the apical-basal axis of late endosomes in zone 2, cofilin was slightly abundant in the apical region, whereas the phalloidin signal was stronger in the basal region (*Figure 9D*). Note that the possibility that overexpression of YFP-cofilin changed the distribution of actin filaments cannot be excluded. EGFP-actin on a late endosome showed that treatment of E7.5 whole embryos with the S3 peptide (15 µg/ml) for 24 h resulted in a decrease in radially extended actin filaments in the apical area of zone 2, whereas the actin filaments extending along the

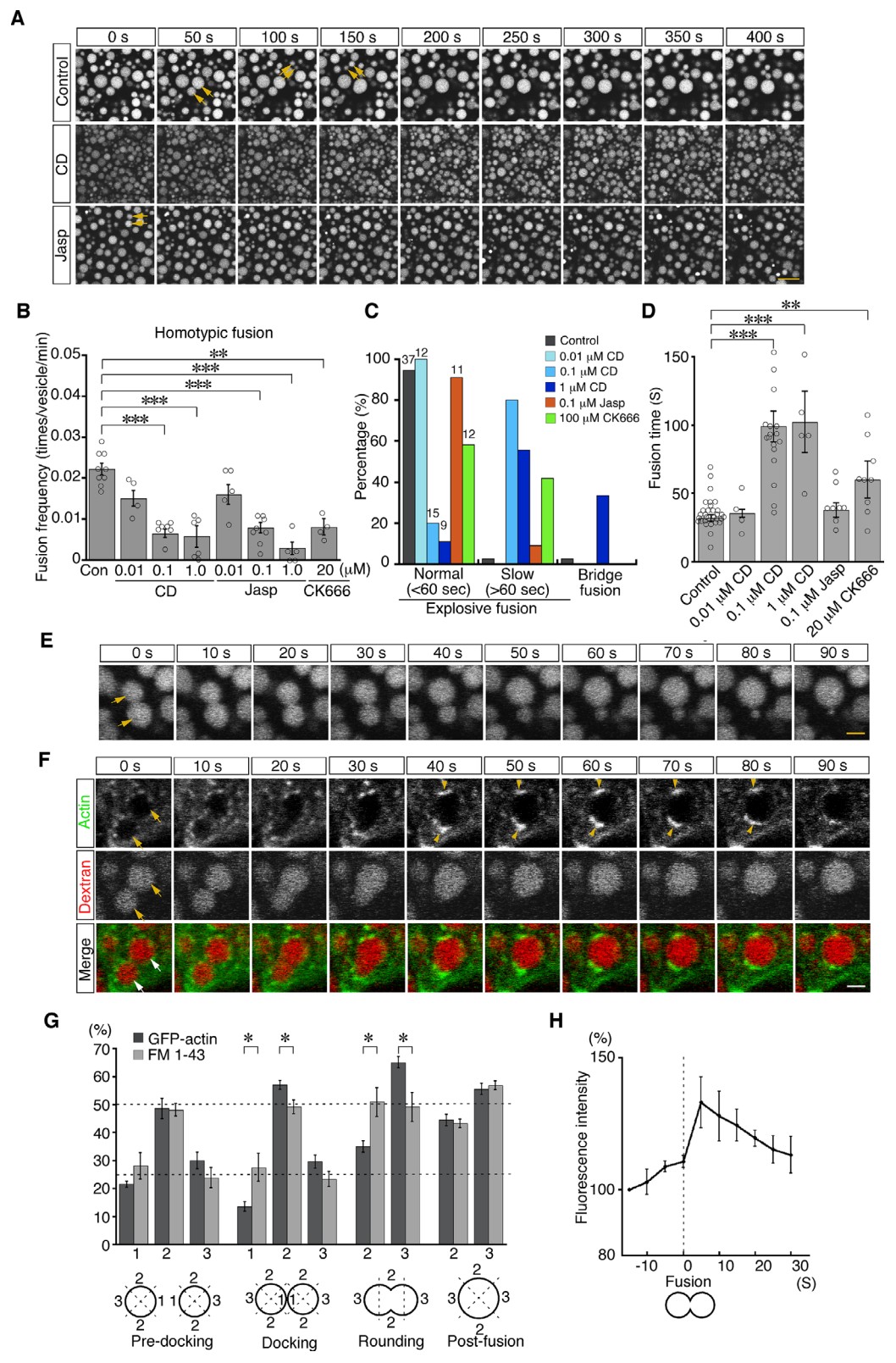

**Figure 7.** Actin dynamics regulate homotypic fusion in VE cells. (**A**) Confocal microscopic images of endocytic vesicles in zone 2. E8.5 mouse embryos pretreated with 0.1 μM cytochalasin D (CD) or 0.1 μM jasplakinolide (Jasp) were labeled with Alexa 488-transferrin for 5 min. Time-lapse images were taken for 400 s from 5 min after labeling. Arrows indicate homotypic fusion. Time 0 indicates the start of time-lapse imaging. (**B**) Frequency of

*Figure 7 continued on next page*

*Figure 7 continued*

homotypic fusion calculated from the number of fusions that occurred in 400 s. Data are represented as means ± SEMs of four to nine independent experiments; individual data are shown by circular dots. Data were analyzed by use of one-way ANOVA with the Tukey multiple comparison test (**p<0.01; ***p<0.001). (**C**) Quantitation of types of homotypic fusion. Fusion types are classified into normal explosive fusion, slow explosive fusion (more than 60 s), and bridge fusion. Cytochalasin D treatment slowed down the fusion process in a dose-dependent manner and induced bridge fusion at 1 μM. Independent experiment numbers are shown on the top of the bars. (**D**) Quantitation of the time for completion of homotypic fusion. Treatment with 0.1 μM or 1 μM cytochalasin D or with 20 μM CK666, but not with 0.1 μM jasplakinolide, resulted in a significant increase in the fusion time. Data are represented as means ± SEMs of 4–30 independent experiments; individual data are shown by circular dots. Data were analyzed by use of one-way ANOVA with the Tukey multiple comparison test (**p<0.01; ***p<0.001). (**E**) Representative images of homotypic fusion of late endosomes in VE cells treated with 1 μM cytochalasin D. Endosomes were visualized by incubation with Alexa 488-transferrin. The arrows indicate the vesicles undergoing homotypic fusion in the bridge fusion mode. Time 0 refers to the start of fusion of the endosomes indicated by arrows. (**F**) Time-lapse imaging of EGFP-actin localization during homotypic fusion of late endosomes. Localization of EGFP-actin (green) and endocytic vesicles (red), visualized by use of rhodamine-dextran, was observed. When a pair of endosomes (arrows) fused, EGFP-actin was newly polymerized at the opposite positions of the docking site and its fluorescence intensity was increased (arrowheads). Time 0 refers to the start of fusion of the endosomes indicated by arrows. (**G**) Distribution of EGFP-actin during homotypic fusion of late endosomes. The fluorescence intensities of EGFP-actin (dark gray, n=5) and FM 1–43 (light gray, n=5) were measured during homotypic fusion. The percentages of their signals in the proximal (docking site, shown as 1), lateral (shown as 2), and distal (shown as 3) regions at the predocking, docking, rounding, and postfusion stages are shown. Data are represented as means ± SEMs (*p<0.05, unpaired t-test). (**H**) The time course of changes in the fluorescence intensity of EGFP-actin around a pair of late endosomes during homotypic fusion. Time 0 s indicates the moment the fusion pore of two late endosomes was formed. The graph indicates the average of five independent experiments. The scale bars indicate 10 μm (**A**), 2 μm (**E**), and 3 μm (**F**).

The online version of this article includes the following figure supplement(s) for figure 7:

**Figure supplement 1.** Distribution of actin filaments and homotypic fusion in cytochalasin D-treated VE cells.

**Figure supplement 2.** Distribution of actin filaments in jasplakinolide-treated VE cells.

**Figure supplement 3.** Distribution of actin during homotypic fusion in cytochalasin D-treated VE cells.

**Figure supplement 4.** Distribution of Arp3-EGFP during fusion.

---

apical-basal axis appeared unchanged (*Figure 9—figure supplement 1*). In addition, a FRAP assay showed that the treatment of S3 peptide induced faster turnover of actin filaments of late endosomes when compared with treatment with the RV peptide (*Aizawa et al., 2001*), which contains the reverse sequence of cofilin and is used as a negative control ($t_{1/2}$ = 10.9 s with the S3 peptide, n=9 vs $t_{1/2}$ = 18.3

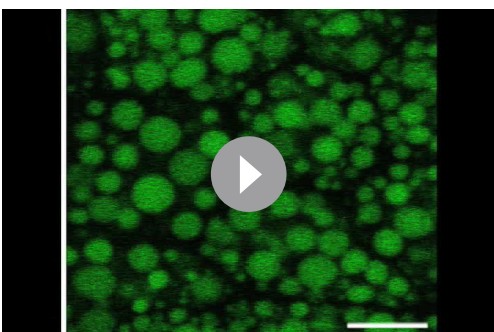

**Video 3.** Homotypic fusion between late endosomes in VE cells treated with 0.1 μM cytochalasin D. Time-lapse recording of homotypic fusion of late endosomes. VE cells pretreated with 0.1 μM cytochalasin D for 5 min were labeled with Alexa Fluor 488-transferrin. Homotypic fusion of late endosomes was observed. The scale bar indicates 10 μm.

https://elifesciences.org/articles/95999/figures#video3

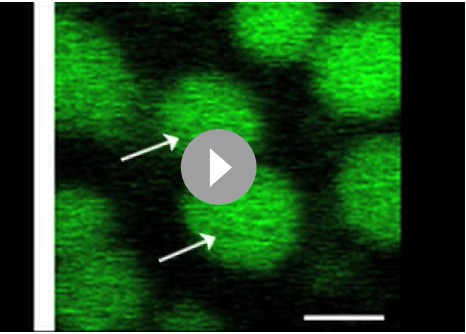

**Video 4.** Bridge fusion between late endosomes in VE cells treated with 1 μM cytochalasin D. Time-lapse recording of homotypic fusion of late endosomes. VE cells pretreated with 1 μM cytochalasin D for 5 min were labeled with Alexa Fluor 488-transferrin. Homotypic fusion of late endosomes was observed. The scale bar indicates 3 μm.

https://elifesciences.org/articles/95999/figures#video4

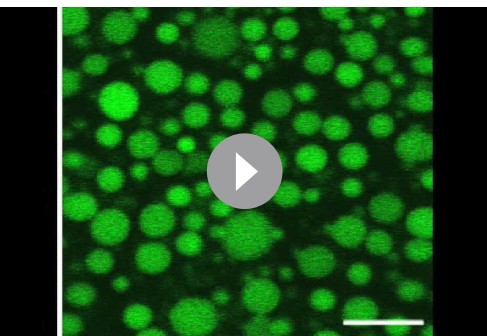

**Video 5.** Homotypic fusion between late endosomes in VE cells treated with 1 μM jasplakinolide. Time-lapse recording of homotypic fusion of late endosomes. VE cells pretreated with 1 μM jasplakinolide for 5 min were labeled with Alexa Fluor 488-transferrin. Homotypic fusion of late endosomes was observed. The scale bar indicates 10 μm.

https://elifesciences.org/articles/95999/figures#video5

s with the RV peptide, n=9; *Figure 9E*). These results indicate that cofilin distributes with actin filaments on late endosomes and regulates actin dynamics on late endosomes. Finally, the roles of cofilin in late endosomal fusion were investigated. After treatment with the S3 peptide for 24 hr, Alexa 488-transferrin-labeled vesicles were observed by use of time-lapse imaging. When compared with the RV peptide, the S3 peptide strongly inhibited homotypic fusion (*Figure 9F*) but not heterotypic fusion (*Figure 9G*). These data suggest that spatial differentiation of actin dynamics along the apical-basal direction, partially contributed to by cofilin, may be a critical factor for the transition of fusion modes (*Figure 10*).

## Discussion

In this study, we developed a novel imaging method of analyzing the motile behavior of late endosomes in VE cells and found that VE cells have two different types of vesicle fusion regulated by distinct actin-mediated mechanisms.

### Actin filaments are associated with endosomal vesicles

We found two types of actin filaments associated with late endosomes in VE cells: those that extended along the apical-basal axis and those that extended radially from the endosomes (*Figure 5*). Our findings resemble the previous observation that early endosomes were distributed along with actin cables and that Arp2/3-dependent small cloud-like patches protruded from the membrane (*Morel et al., 2009*). Actin is reportedly distributed around vesicles in different patterns. In addition to the well-known interaction of vesicles with actin rails in directional vesicle transport (*Titus, 2018*), actin was localized at the vertex ring in vacuole fusion in yeast (*Wang et al., 2002*) and entirely surrounded phagosomes in macrophages (*Liebl and Griffiths, 2009*). Advanced microscopy techniques have clarified the fine structure of actin, which is assembled into polyhedron-like lattices around vesicles in exocrine glands (*Ebrahim et al., 2019*).

We observed strong actin signals as patches on late endosomes in the focal planes on confocal microscopy. Three-dimensional reconstruction showed that most of these patches represent the cross-section of rod-shaped actin filaments extending from the apical side into the cytoplasm in VE cells. In addition to these patches, some signals seem to represent actin filaments extending radially from membranes. Furthermore, we showed that Arp3-positive actin spots appeared in the site opposite to that of endosome movement (*Figure 6F*, *Figure 6—figure supplement 1C*) like a comet tail (*Goley and Welch, 2006*), that Arp3-positive actin spots were asymmetrically distributed and temporarily polymerized in the distal region during endosomal fusion (*Figure 7F and G*; *Figure 7—figure supplement 4*), and that Arp2/3 inhibition reduced homotypic fusion (*Figure 7B*). These results suggest that radial actin filaments polymerized by Arp2/3 generate force for homotypic fusion in VE cells. This notion is compatible with the findings of previous reports that the radial filaments were nucleated and extended with Arp2/3-dependent branches on the lateral membrane of endosomes to push vesicles (*Goley and Welch, 2006*; *Svitkina, 2018*). Given that nucleation and Arp2/3 activities are regulated by nucleation-promoting factors (NPFs; *Goley and Welch, 2006*), the activity of NPFs and membrane fusion processes are likely tightly linked to accurate membrane fusion. Differential regulation of NPF activity may give rise to different patterns of actin surrounding vesicles and membrane dynamics. Future investigation is necessary to elucidate how the NPF activity is spatially and temporally regulated on membranes.

Actin filaments distribute on endosomes in various patterns depending on the cell types. However, why cells show different patterns of actin filament distribution on vesicles and whether different

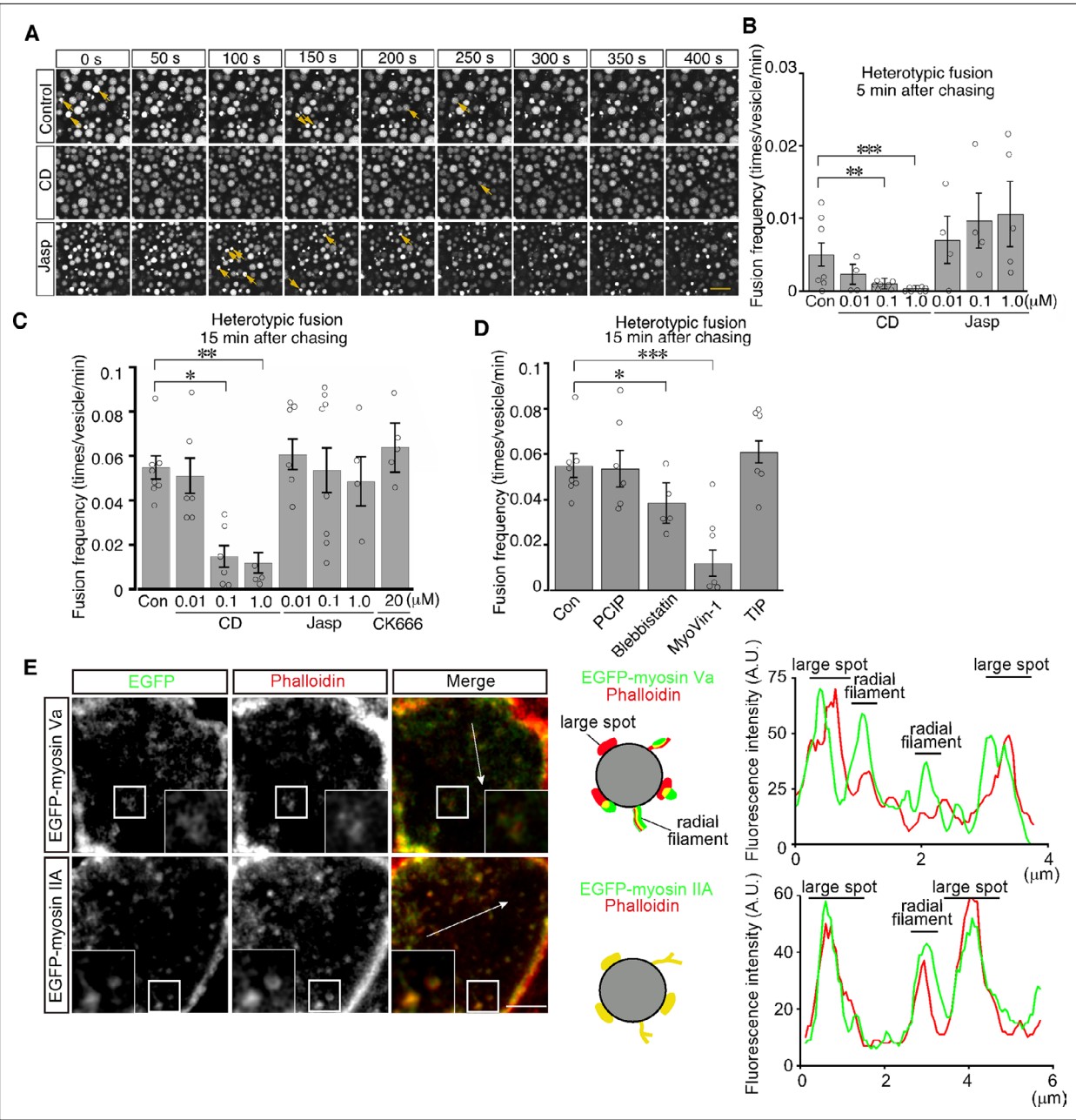

**Figure 8.** Actin stabilization induces heterotypic fusion of late endosomes with lysosomes. (**A**) Confocal microscopic images of endocytic vesicles in zone 2. E8.5 mouse embryos pretreated with 0.1 μM cytochalasin D (CD) or with 0.01 μM jasplakinolide (Jasp) were labeled with Alexa 488-transferrin for 5 min. Time-lapse images were taken for 400 s from 15 min after labeling. The arrows indicate heterotypic fusion with lysosomes. Time 0 indicates the start of time-lapse imaging. (**B**) Frequency of heterotypic fusion calculated from the number of fusions that occurred in 400 s from 5 min after labeling. Data are represented as means ± SEMs of four to seven independent experiments; individual data are shown by circular dots. Data were analyzed by use of one-way ANOVA with the Tukey multiple comparison test (**p<0.01, ***p<0.001). (**C**) Quantitation of heterotypic fusion frequency at 15 min after labeling of VE cells pretreated with cytochalasin D (CD), jasplakinolide (Jasp), or CK666. Data are represented as means ± SEMs of four to eight independent experiments; individual data are shown by circular dots. Data were analyzed by use of one-way ANOVA with the Tukey multiple comparison test (*p<0.05, **p<0.01). (**D**) Effects of myosins on heterotypic fusion. Frequency of heterotypic fusion was calculated at 15 min after labeling. Data are represented as means ± SEMs of five to eight independent experiments; individual data are shown by circular dots. Data were analyzed by use of one-way ANOVA with the Tukey multiple comparison test (*p<0.05, ***p<0.001). (**E**) Colocalization of EGFP-myosin IIA or EGFP-myosin Va with phalloidin in zone 2 of VE cells. The insets show the magnified images of the boxed area. Following electroporation of the EGFP-fused myosin expression plasmids and 24 hr culture, whole embryos were stained with Alexa 546-phalloidin. EGFP-myosin IIA signals overlapped with phalloidin-positive spots on late endosomes, whereas most of the EGFP-myosin Va signals were juxtaposed with phalloidin-positive spots on late endosomes (see the schemas and fluorescence intensity plots). The scale bars indicate 10 μm (**A**), 3 μm (**E**), and 1 μm (E, insert).

*Figure 8 continued on next page*

*Figure 8 continued*

The online version of this article includes the following figure supplement(s) for figure 8:

**Figure supplement 1.** Effects of myosin inhibitors on homotypic fusion in VE cells.

distribution patterns of actin filaments have different functions remain unclear. In this study, by taking advantage of the large size of endosomes, we found that there are two different types of actin filament distribution on the surface of late endosomes in VE cells and that these actin filaments regulate different aspects of membrane fusion, movement, and endosomal distribution. Taken together, these data suggest that different patterns of actin filament distribution on vesicles have various effects on specific functions of vesicles.

## Actin regulates membrane behavior after pore formation

In this study, we found that the membranes of late endosomes behave differently depending on the fusion mode. In homotypic fusion, the fusion pore formed between docked endosomal vesicles expanded quickly and the fusion was completed within 20-30 s (explosive fusion, *Figure 2*). In contrast, in heterotypic fusion with lysosomes, the pore opened very slowly and late endosomes gradually shrank in diameter and disappeared over 1-2 min (bridge fusion, *Figure 3*). Using mathematical modeling, we showed that vesicle size is an important factor determining the fusion mode. When the size of the vesicle becomes larger, the free energy for neck relaxation decreases and the pore is kept small. Our modeling predicted that the late endosomes are large enough to allow bridge fusion to occur, although they exhibit explosive fusion in vivo. Similarly, homotypic fusion in an explosive mode was observed in very large vesicles, such as vacuoles in yeast, endosomal vesicles in coelomocytes of *C. elegans*, and secretory granules in the salivary glands (*Liu et al., 2016*; *Masedunskas et al., 2011*; *Wang et al., 2002*). These observations indicate that vesicle size is not the sole factor determining the fusion mode. Lipid composition, a well-known parameter affecting its bending modulus, is also involved, but the effect on fusion mode may be less dominant when compared with the vesicle size effects (*Simson et al., 1998*; *Faizi et al., 2019*; *Ridolfi et al., 2021*). One of the critical factors other than vesicle size that determine the fusion mode is membrane fluctuation. We found that homotypic fusion was converted to bridge-like fusion after cytochalasin D treatment (*Figure 7C*), which suggests that actin dynamics around vesicles cause fluctuation and affect the fusion modes. This finding can be successfully explained by introducing minor modifications in our mathematical modeling so that a larger fluctuation can shift the fusion mode from bridge to explosive fusion.

VE cells vigorously endocytose maternal proteins and supply the digested products to the developing embryo to support embryonic development. This seems to be why VE cells have such huge

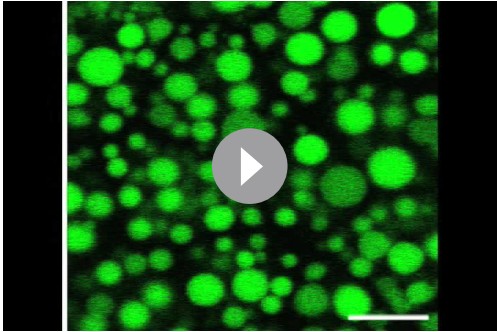

**Video 6.** Heterotypic fusion between late endosomes in VE cells treated with 0.1 µM cytochalasin D. Time-lapse recording of heterotypic fusion between late endosomes and lysosomes in VE cells treated with 0.1 µM cytochalasin D. At 15 min after labeling with Alexa Fluor 488-transferrin, VE cells of embryos were observed. The scale bar indicates 10 µm.

https://elifesciences.org/articles/95999/figures#video6

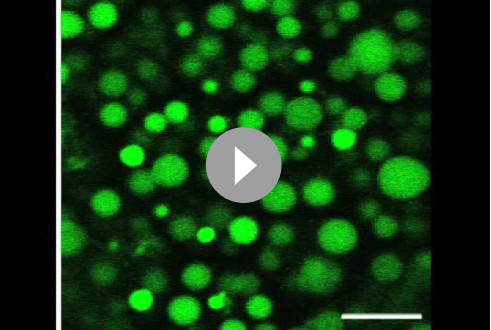

**Video 7.** Heterotypic fusion between late endosomes in VE cells treated with 1 µM jasplakinolide. Time-lapse recording of heterotypic fusion between late endosomes and lysosomes in VE cells treated with 1 µM jasplakinolide. At 15 min after labeling with Alexa Fluor 488-transferrin, VE cells of embryos were observed. The scale bar indicates 10 µm.

https://elifesciences.org/articles/95999/figures#video7

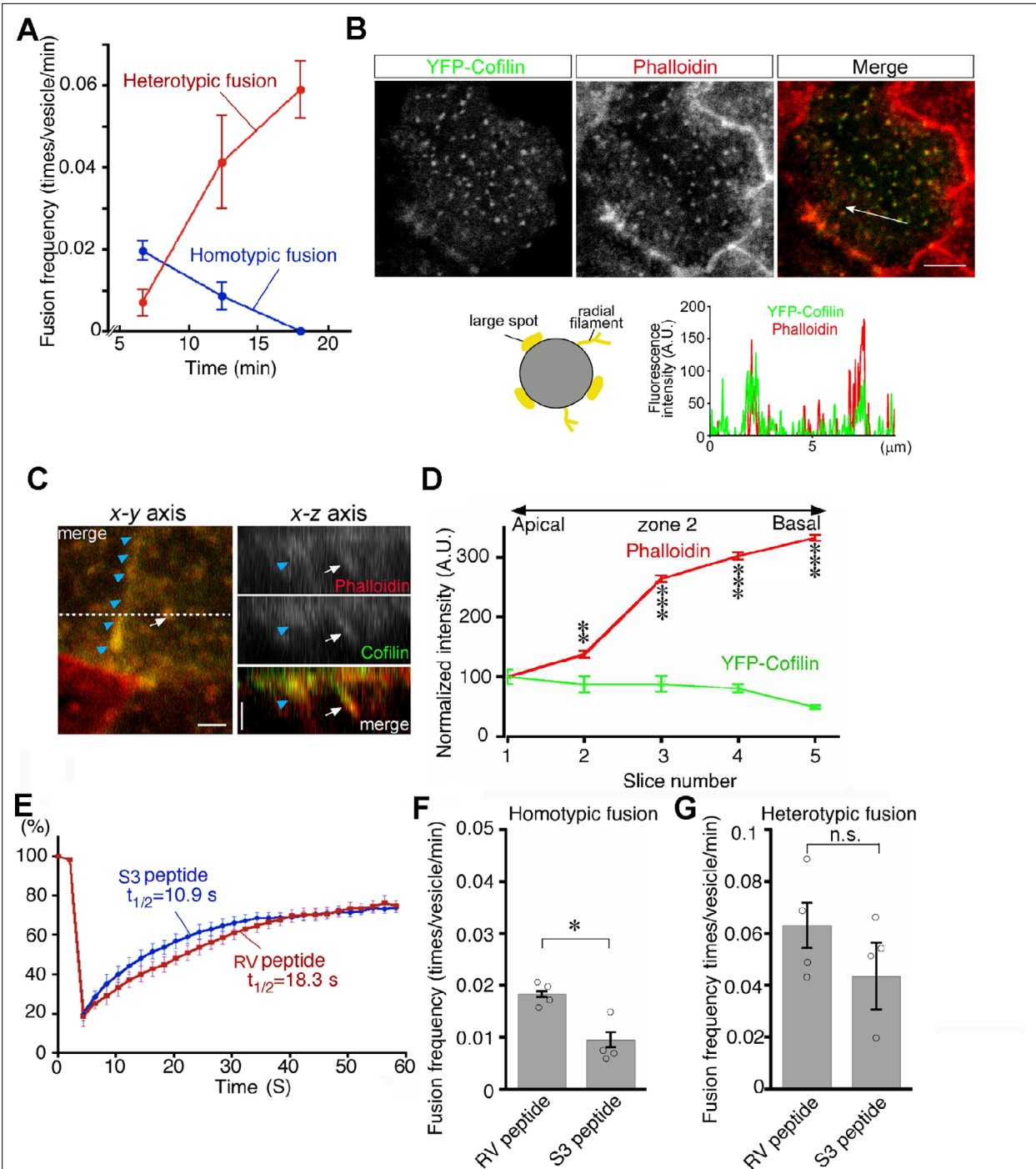

**Figure 9.** Transition from homotypic to heterotypic fusion. (**A**) Time course of frequencies of homotypic (blue line) and heterotypic (red line) fusion. The graph indicates the means of 6 independent experiments. (**B**) Distribution of cofilin in zone 2 of VE cells. After whole embryos were electroporated with YFP-cofilin and cultured for 1 d, they were stained with Alexa 546-phalloidin. The lower graph plotting the fluorescence intensity along the white arrow in the upper image indicates colocalization of actin filaments and cofilin. (**C**) Optical sectioning images (right) of VE cells along the dotted line in the x-y axial panel (left). Embryos, which were electroporated with YFP-cofilin and cultured for 1 d, were stained with Alexa 546-phalloidin. The actin filaments, extending from the apical to the basal region, were highly colocalized with cofilin. Note that the white arrows and blue arrowheads indicate an actin filament that surrounds late endosomes and the cell membrane, respectively. (**D**) Quantification of the fluorescence intensity of phalloidin and cofilin along the apical-basal axis. The intensity of spots colocalized with phalloidin and cofilin around late endosomes observed in the x-y images was quantified along the z-axis. Fluorescence intensity was normalized to the intensities in the most apical plane (8 late endosomes from 3 VE cells). Data were analyzed by use of one-way ANOVA with the Tukey multiple comparison test and represented as means ± SEMs (**p<0.01, ***p<0.001).

*Figure 9 continued on next page*

*Figure 9 continued*

(**E**) FRAP assay to evaluate the dynamics of actin at the apical region of zone 2 in VE cell pretreated with the S3 peptide. After treatment with the S3 or RV peptide overnight, the area (4 µm x 4 µm) covering a single late endosome in VE cells expressing EGFP-actin was bleached and the fluorescence recovery was observed by time-lapse microscopy. Three independent experiments were performed. (**F, G**) Frequency of homotypic (**F**) and heterotypic fusion (**G**) in embryos pretreated with 15 µg/ml S3 or RV peptides. Frequencies of homotypic and heterotypic fusions were calculated according to the numbers of fusions in 400 s from 5 min (**F**) and 15 min (**G**) after labeling with Alexa 488-transferrin. Data are represented as means ± SEMs of three to four independent experiments; individual data are shown by circular dots. Data were analyzed by use of one-way ANOVA with the Tukey multiple comparison test (*$p<0.05$). The scale bars indicate 8 µm (**B**) and 2 µm (**C**). n.s. indicates no significant difference.

The online version of this article includes the following figure supplement(s) for figure 9:

**Figure supplement 1.** Distribution of actin filaments in S3 peptide-treated VE cells.

endosomal and lysosomal vesicles. However, larger vesicle size leads to the reduction in the vesicle fusion speed and subsequent nutrient trafficking efficiency. If vesicle size were the only factor determining the fusion mode, this disadvantage would occur, but if fluctuation is taken into account, it can be overcome, as shown in our mathematical modeling, and the forces generated by actin are thought to be responsible for increasing the rate of vesicular trafficking. This idea is supported by previous reports that knockout mice of actin-related genes (e.g. *Arc* and *Myh9*) showed abnormal VE cells and embryonic lethality (*Conti et al., 2004*; *Liu et al., 2000*).

In heterotypic fusion, late endosomes fused with lysosomes in a bridge-like manner. This may be because lysosomes are too large in comparison with late endosomes and actin accumulation on the lysosomal membrane is scarce, resulting in insufficient force of actin for explosive fusion. Bridge fusion was previously observed in live cells only when giant vesicles were artificially generated by gene modification, for example, in endosomes enlarged by expression of constitutively active Rab5 (*Roberts et al., 1999*) and lysosomes enlarged in the intestinal cells of *ppk-3*-deleted *C. elegans* (*Nicot et al., 2006*). These findings suggest that the enlargement of these vesicles exceeds the intrinsic forces of actin to induce explosive fusion.

Like homotypic fusion, heterotypic fusion is also dependent on actin, as shown by reduction in fusion frequency upon cytochalasin D treatment. Intriguingly, in heterotypic fusion, NM II, myosin V, and stabilized actin filaments, but not Arp2/3, were required, indicating a distinct contribution of actin to heterotypic fusion when compared with homotypic fusion. We showed that, consistent with the well-known roles of myosin V in trafficking of lysosomes, ER, melanosomes, and synaptic vesicles (*Hammer and Sellers, 2011*), myosin V facilitates the directional transport of late endosomes towards lysosomes along stabilized actin filaments that serve as a rail (*Figure 10*). Moreover, our results indicate that NM II, which is primarily responsible for generating contractile force (*Svitkina, 2018*), partially contributes to heterotypic fusion. It is well established that NM II facilitates the constriction of the membrane into the plasma membrane after membrane pore formation in exocytosis (*Ebrahim et al., 2019*) in cooperation with actin depolymerization by cofilin-1 (*Miklavc et al., 2015*). In contrast, our data showed that actomyosin appeared not to form a lattice on the surface of endosomes for the contraction and that cofilin did not show a robust influence on heterotypic fusion. Rather, actin stabilization tended to promote heterotypic fusion (*Figure 8A and B*). These data suggest that NM II may contribute to heterotypic fusion through a different mechanism from that of exocytosis. NM II is also implicated in the movement of the nucleus along with cell migration, which is akin to a 'conveyor-belt' by the sliding of the nuclear membrane along the actin filament (*Gomes et al., 2005*). Further investigation is necessary to elucidate the mechanisms by which NM II transports late endosomes along actin filaments.

## Determination of fusion targets

Late endosomes, which fuse frequently with neighboring late endosomes, shift the target of fusion towards lysosomes. What triggers the transition of the fusion targets? We found that larger endosomes tend to fuse with lysosomes (*Figure 3G*). Thus, one possibility is that the larger endosomes located in the basal position have a higher opportunity to fuse with lysosomes than do the smaller endosomes in the more apical portion. However, upon treatment with jasplakinolide, even smaller endosomes distributing in the more apical area underwent heterotypic fusion with lysosomes (*Figure 8A and B*), thus suggesting rather that the size and the position of vesicles are not the sole factors and that the different dynamics of actin shift the fusion modes. Furthermore, we showed weaker actin

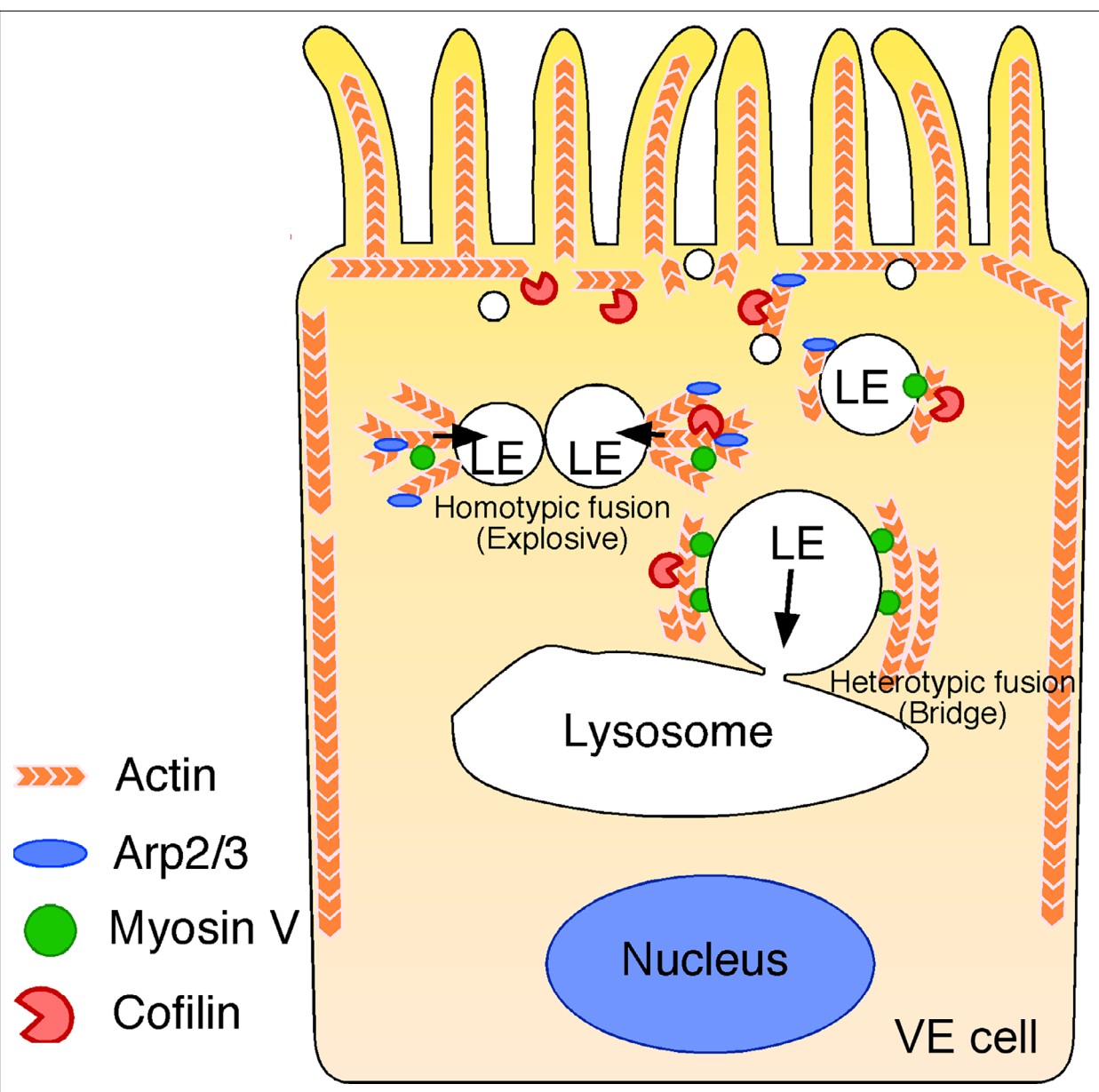

**Figure 10.** Summary of late endosome fusion in VE cells. In VE cells, late endosomes (LE) show two different types of fusion with different targets. In homotypic fusion, once a membrane pore is formed between two late endosomes, the pore quickly expands and a single fused vesicle is formed. In contrast, in heterotypic fusion, the pore does not expand; instead, the late endosome gradually shrinks and disappears over time as the result of a transfer of its vesicle content into lysosomes. Two different types of actin are associated with the late endosomal membrane in VE cells: the actin filaments that extend along the apical-to-basal axis and the filaments that are radially polymerized from the membrane. Cofilin, an actin-binding protein that severs and depolymerizes actin filaments, activates actin turnover. In the apical region, late endosomes receive squeezing forces via active actin modification by cofilin and Arp2/3, which leads to homotypic fusion in an explosive mode. In contrast, in the basal region, actin is more static and late endosomes receive sliding forces via myosins.

polymerization/depolymerization activity in the basal area as compared with that in the apical area (*Figure 6C*), a decrease in the frequency of homotypic fusion but not of heterotypic fusion induced by S3 peptide treatment (*Figure 9F and G*), a decrease in radially extending actin filaments by S3 peptide treatment (*Figure 9—figure supplement 1*), and an increase in actin filaments on endosomes in the basal area as compared with in the apical area (*Figure 9D*). These findings support the notion that cofilin-dependent regulation of actin polymerization/depolymerization may play a critical role in determination of the fusion target. In parallel with VE cells, it was shown that cofilin was specifically distributed in the apical area of epithelial cells and was highly active under the plasma membrane

(*Morris et al., 2011*; *Vitriol et al., 2013*). Cofilin regulates actin remodeling for lamellipodia formation and endosomal trafficking (*Harrington et al., 2011*; *Vitriol et al., 2013*). Furthermore, early endosomes located close to the plasma membrane fuse frequently when compared with late endosomes located at the perinuclear region (*Gruenberg et al., 1989*). Taken together, higher dynamics of actin filaments regulated by cofilin at the cell peripheral area may increase the fusion frequency in many cells.

In this study, we developed a method of observing and manipulating late endosomes in VE cells. This method enables, for the first time, the observation of fusion of single vesicles with two different targets and is very useful to examine the molecular mechanisms that determine the fusion targets of vesicles and regulate different fusion modes.

# Materials and methods
## Key resources table

| Reagent type (species) or resource | Designation | Source or reference | Identifiers | Additional information |
|---|---|---|---|---|
| Gene (*M. musculus*) | Fascin-1 | NCBI Reference Sequence | L33726.1 | |
| Gene (*M. musculus*) | myosin Va | NCBI Reference Sequence | XM 006510827.5 | |
| Gene (*M. musculus*) | myosin IIA | NCBI Reference Sequence | NM 022410.4 | |
| Gene (*M. musculus*) | Arp3 | NCBI Reference Sequence | AK004554.1 | |
| Recombinant DNA reagent (plasmid) | pEGFP-Rab7 | doi: 10.1038 /s41467-019-09617-9 | | Backbone:pEGFP C2 vector |
| Recombinant DNA reagent (plasmid) | pmRFP-Fascin-1 | This paper | | Backbone:pmRFP C1 vector |
| Recombinant DNA reagent (plasmid) | pEGFP-myosin Va | This paper | | Backbone:pEGFP C2 vector |
| Recombinant DNA reagent (plasmid) | pEGFP-myosin IIA | This paper | | Backbone:pEGFP C2 vector |
| Recombinant DNA reagent (plasmid) | pYFP-Cofilin | Kindly provided by Kensaku Mizuno | | Backbone:pEGFP C1 vector |
| Recombinant DNA reagent (plasmid) | pEGFP-actin | BD Biosciences | #6116–1 | Backbone:pEGFP C1 vector |
| Recombinant DNA reagent (plasmid) | Arp3-pEGFP-N | This paper | | Backbone:pEGFP N1 vector |
| Antibody | Mouse anti-EEA1 monoclonal antibody, FITC conjugated, clone 14 | BD Biosciences | AB_399409 | 1:100 for IF |
| Antibody | Cy3-AffiniPure goat anti-mouse IgG (H+L) | Jackson ImmunoResearch | AB_2338690 | 1:500 for IF |
| Strain, strain background (*M. musculus*) | ICR/CD1 | CLEA Japan, Japan SLC | | |
| Chemical Compound, drug | FM 1–43 Dye | Molecular Probes | T3163 | |
| Chemical Compound, drug | LysoTracker Red DND-99 | Molecular Probes | L7528 | |
| Chemical Compound, drug | Transferrin, Alexa Fluor 488 conjugate | Molecular Probes | T13342 | |
| Chemical Compound, drug | Transferrin, Alexa Fluor 594 conjugate | Molecular Probes | T13343 | |
| Chemical Compound, drug | Dextran, Rhodamin B, 70,000 MW | Molecular Probes | D1841 | |
| Chemical Compound, drug | Alexa Fluor 488-phalloidin | Molecular Probes | A12379 | |
| Chemical Compound, drug | Alexa Fluor 546-phalloidin | Molecular Probes | A22283 | |
| Chemical Compound, drug | Cytochalasin D | Sigma-Aldrich | C8273 | |
| Chemical Compound, drug | Jasplakinolide | Calbiochem | 420107 | |

*Continued on next page*

Continued

| Reagent type (species) or resource | Designation | Source or reference | Identifiers | Additional information |
|---|---|---|---|---|
| Chemical Compound, drug | CK666 | Calbiochem | 182515 | |
| Chemical Compound, drug | Pentachloropseudilin | AOBIOUS | AOB33969 | |
| Chemical Compound, drug | Blebbistatin | Sigma-Aldrich | 203389 | |
| Chemical Compound, drug | MyoVin-1 | Calbiochem | 475984 | |
| Chemical Compound, drug | 2,4,6-triiodophenol | Tokyo Chemical Industry | T0452 | |
| Software, algorithm | Fiji-Image J | https://imagej.net/software/fiji/downloads | | |
| Software, algorithm | Illustrator | Adobe | | |
| Software, algorithm | Photoshop | Adobe | | |
| Software, algorithm | Canvas for Mac | Canvas GFX | | |
| Software, algorithm | KaleidaGraph | SYNERGY SOFTWARE | | |

## Animal experiments

All the experiments using animals were approved by the Animal Care and Use Committee of the University of Tsukuba (#24–248) and of the University of Toyama (A2021ENG-1) and performed under their guidelines. Noon of the day on which a vaginal plug was observed was taken as embryonic day 0.5 (E0.5). Time-pregnant ICR mice were purchased from CLEA Japan and Japan SLC.

## Immunohistochemistry and phalloidin staining

For whole-mount immunohistochemistry, E8.5 embryos were fixed with 4% paraformaldehyde (PFA) in PBS at 4 °C for 3 hr and incubated with FITC-conjugated anti-EEA1 (1:100, BD Bioscience) or Cy3-labeled anti-mouse IgG (1:500, Jackson ImmunoResearch). For phalloidin staining, the embryos were fixed with 4% PFA in PBS at 37 °C for 15 min and incubated with 0.2 U/ml Alexa Fluor 488- or Alexa Fluor 546-conjugated phalloidin (Molecular Probes) for 60 min.

## Ex vivo whole embryo culture

The whole embryo culture was performed as described previously (*Koike et al., 2009*). In brief, embryos were dissected at E7.5 and cultured in 100% rat serum (Charles River) supplemented with 2 mg/ml glucose in a culture bottle placed in a rotation drum culture system (Ikemoto Rika) at 37 °C under 5% $O_2$, 5% $CO_2$, and 90% $N_2$ for 24 hr. In the pharmacologic experiments, embryos were incubated with 15 µg/ml S3 and RV peptides (a kind gift of Dr Kensaku Mizuno, Tohoku University).

## Endocytic uptake assays and intracellular vesicle tracing

Whole embryos with intact yolk sacs dissected out at E8.5 or embryos cultured for 1 d from E7.5 were transferred into DMEM (Sigma-Aldrich) supplemented with 0.1% BSA. Embryos were incubated in 100 nM LysoTracker Red (Molecular Probes) or 12 µM FM 1–43 (Molecular Probes) in 0.1% BSA/DMEM at 37 °C for 5 min. For tracing endocytic vesicles, embryos were incubated with 20 µg/ml Alexa Fluor 488-labeled transferrin (Molecular Probes) or 100 µg/ml dextran rhodamine B 70,000 MW (Molecular Probes) at 37 °C for 5 min. For differential labeling of late endosomes and lysosomes, whole embryos were initially cultured in 100% rat serum supplemented with 2 mg/ml glucose and 75 µg/ml dextran rhodamine B 70,000 MW at 37 °C for 20 min and then incubated with 20 µg/ml Alexa Fluor 488-labeled transferrin for 5 min. To test the effects of actin inhibitors, embryos were incubated with 0.01–1 µM cytochalasin D (Sigma-Aldrich), 0.01–1 µM jasplakinolide (Calbiochem), 20 µM CK666 (Calbiochem), 10 µM pentachloropseudilin (AOBIOUS), 10 µM blebbistatin (Sigma-Aldrich), 10 µM MyoVin-1 (Calbiochem), and 10 µM 2,4,6-triiodophenol (Tokyo Chemical Industry) for 5 min, and the same drugs were added to the medium during observation.

## Image acquisition and analysis

For high-resolution imaging, the samples on the glass-bottom dish were mounted with ProLong Glass (Thermo Fisher Scientific). Images were captured by use of a TCS SP8 laser confocal microscope (Leica Microsystems) equipped with an HC PL APO CS2 100×/1.40 OIL objective (Leica). Alexa Fluor 488 and Alexa Fluor 594 dyes were excited at 488 and 594 nm of a supercontinuum white light laser, respectively. The emitted fluorescence was sequentially detected by use of HyD-SMD detectors through a spectral separation module (505–550 nm for Alexa Fluor 488 and 605–700 nm for Alexa Fluor 594). The pinhole size was set at 0.40 Airy units (60.7 μm). The pixel size and z-step were set at 45 nm and 182 nm, respectively. The acquired images were processed using the Lightning deconvolution algorithm on LAS-X software (Leica). Three-dimensional reconstructions of the images were carried out on NIS-Elements software (Nikon) or Imaris software (Bitplane).

An embryo was placed on a glass-bottom dish (No.1S thickness, Matsunami Glass), and confocal fluorescent images were obtained by use of an LSM510 laser scanning microscope (Carl Zeiss) equipped with a C-APOCHROMAT 63 x/1.2 W Korr objective lens (Carl Zeiss) and a pinhole set to 1.0 AU. Images of zone 2 and zone 3 were taken by focusing on the z-axis where late endosomes and lysosomes occupied the largest area in the observation field, respectively. To compare endosomal sizes, the area of all intact endosomes in each image was measured by use of Image J. Time-lapse images were obtained by collecting 40 images with time intervals of 5 or 10 s at room temperature. The series of images of the confocal z-stacks were obtained with a lateral resolution of 1024x1024 pixels and a scanning depth of 6–8 μm with 13–17 layers and 0.5 μm intervals. To quantify fusion frequency, the number of membrane fusion events in the 400 s time-lapse images was counted, and the number of fusions per minute a single late endosome underwent was calculated. To compare the time required for single fusion processes, fusion time was defined as follows. For homotypic fusion, fusion time was defined as the time from the start of membrane fusion of two late endosomes until the formation of a single, round vesicle. For heterotypic fusion, fusion time was defined as the time from when the late endosomes began to shrink until they disappeared as the result of absorption by a lysosome.

## Electroporation of whole embryos

The electroporation of whole embryos was described previously (*Koike et al., 2009*). An E7.5 embryo was inserted into a small hole of an agarose mold (2% agarose/PBS, 7x7 mm) placed in Hanks buffered saline solution (HBSS). The solution bathing the embryos was then replaced with HBSS containing 2 mg/ml DNA. Three square electric pulses (30 V, 50ms duration, 1 pulse/s) were delivered to the agarose mold by use of an electroporator (CUY21, Nepa Gene) and a 10 mm electrode (CUY650P10, Nepa Gene). After electroporation, the embryo was rinsed with HBSS and cultured as described in the previous section. Rab7, nonmuscle myosin IIA, and myosin Va subcloned into a pEGFP-C2 expression vector (Clontech), Arp3 subcloned into a pEGFP-N1 expression vector (Clontech), fascin-1 subcloned into a pmRFP-C1 expression vector (Clontech), pEGFP-Actin (BD Biosciences), and YFP-cofilin (kind gifts of Dr Kensaku Mizuno) subcloned into the pEGFP-C1 vector (Clontech) were used. For cloning, following primers were used: myosin IIA Fwd, 5'-ttcgaattctgcatggctcagcaggctgcag-3'; myosin IIA Rev, 5'-cagaattcctattcagctgccttggcatc-3'; myosin Va Fwd, 5'-ttcgaattctgcatggccgcgtccgagctctac-3'; myosin Va Rev, 5'- cagaattctcagacccgtgcgatgaag-3'; fascin-1 Fwd, 5'-ttcgaattctgcatgaccgccaacggca cg-3'; fascin-1 Rev, 5'-cagaattcctagtactcccagagtgagg-3'; Arp3 Fwd, 5'-tcgaattcatggcgggacggctgc c-3'; and Arp3 Rev, 5'-gtggatcccgggacatgactccaaacactg-3'.

## Photobleaching assay

Photobleaching analysis was performed with the LSM510 laser scanning microscope. The EGFP-actin-electroporated embryo was observed by use of a 63 x objective lens. For the bleaching, an ROI was drawn around a late endosome (4 μm x 4 μm) or cell boundary and was bleached by use of a 488 nm laser beam for 8 bleach iterations at 100% intensity, and fluorescence recovery into the bleached area was monitored. To calculate the percentage of recovery after photobleaching, the intensity profiles for the actin filaments in the bleached area, a background region, and the whole cells were collected. The background was subtracted from all the data, and the intensity of the whole cells was used to correct for fluorescence loss during image acquisition.

## Electron microscopy

E8.5 mouse embryos were fixed with 2% glutaraldehyde and 2% PFA in 0.1 M phosphate buffer (pH 7.4) overnight. After three washes with 0.1 M phosphate buffer, pH 7.4, the embryos were postfixed with 1.0% osmium tetroxide in 0.1 M phosphate buffer for 2 hr, dehydrated, and embedded in Epon 812 according to a standard procedure. Ultrathin sections were stained with uranyl acetate and lead citrate and observed by use of an electron microscope (H-7100, Hitachi).

## Acknowledgements

This work was supported in part by KAKENHI (17024006) and the 21st Century COE program from the MEXT, Japan, KAKENHI (JP22H04926), Grant-in-Aid for Transformative Research Areas – Platforms for Advanced Technologies and Research Resources "Advanced Bioimaging Support" from JSPS, Japan, and Takeda Science Foundation. We thank Drs K Mizuno, K Ohashi, and N Ohsumi, for reagents, protocols, and advice; Y Sato (Carl Zeiss, Japan) for help in 3D reconstruction; and H Bito and F Miyamasu for useful comments on the manuscript.

## Additional information

### Funding

| Funder | Grant reference number | Author |
|---|---|---|
| Ministry of Education, Culture, Sports, Science and Technology | KAKENHI (17024006) | Masayuki Masu |
| Japan Society for the Promotion of Science | KAKENHI (JP22H04926) | Seiichi Koike |
| Takeda Science Foundation | 2022039209 | Seiichi Koike |

The funders had no role in study design, data collection and interpretation, or the decision to submit the work for publication.

### Author contributions

Seiichi Koike, Conceptualization, Formal analysis, Investigation, Visualization, Methodology, Writing - original draft, Writing - review and editing; Masashi Tachikawa, Software, Formal analysis, Visualization, Methodology, Writing - review and editing; Motosuke Tsutsumi, Tomomi Nemoto, Investigation, Visualization, Methodology, Writing - review and editing; Takuya Okada, Kazuko Keino-Masu, Conceptualization, Investigation, Writing - review and editing; Masayuki Masu, Conceptualization, Supervision, Funding acquisition, Investigation, Writing - original draft, Project administration, Writing - review and editing

### Author ORCIDs

Seiichi Koike ⓘ https://orcid.org/0000-0001-8203-4498
Motosuke Tsutsumi ⓘ https://orcid.org/0000-0002-5832-3828
Tomomi Nemoto ⓘ https://orcid.org/0000-0001-6102-1495
Masayuki Masu ⓘ https://orcid.org/0000-0002-4726-7429

### Ethics

All the experiments using animals were approved by the Animal Care and Use Committee of the University of Tsukuba (#24-248) and of the University of Toyama (A2021ENG-1) and performed under their guidelines.

Reviewer #1 (Public review): https://doi.org/10.7554/eLife.95999.3.sa1
Reviewer #3 (Public review): https://doi.org/10.7554/eLife.95999.3.sa2
Author response https://doi.org/10.7554/eLife.95999.3.sa3

# Additional files

## Supplementary files
• MDAR checklist

## Data availability
All data generated or analysed during this study are included in the manuscript and supporting files; there are no source data files.

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

## Appendix 1

### Free energy dependence on the size and shape of membrane

In general, the free energy governing the shape relaxation process of a vesicle (closed lipid membrane) consists of 2 terms: the bending energy and the osmotic energy,

$$F = b - \Delta p \cdot V$$

The bending energy, $b$, is given by

$$b = \frac{\kappa}{2} \int_{surface} c^2 \cdot da,$$

where $c$ is the mean curvature defined at each point of the surface of the vesicle and the integration is performed all over the vesicle. $\kappa$ is the bending modulus (**Helfrich, 1973**). The mean curvature has the dimension of the inverse of the length. Suppose a similarity transformation of a membrane with the length scaling ratio $\lambda$, the mean curvature of the transformed membrane is rescaled by $1/\lambda$ and the surface area is rescaled by $\lambda^2$. Thus, the bending energy is invariant under a scale change in size.

The osmotic energy is the product of the osmotic pressure difference between the inside and the outside of the membrane, $\Delta p$, and the volume enclosed by the membrane, $V$. The volume depends on both the size and the shape of the membrane. Here we represent the size of a vesicle with its area $A$, since it is invariant under shape changes, and we introduce a rescaled volume $v$ ($v = 0 \sim 1/6\sqrt{\pi}$) with

$$V = A\sqrt{A} \cdot v.$$

Thus,

$$F = b - \Delta p \cdot A\sqrt{A} \cdot v.$$

In the above form, a similarity transformation only changes the scale factor term $A\sqrt{A}$. $b$ and $v$ depend only on the shape and are invariant under a scale change in size.

### Description of shape relaxation

In this study, we consider a process in which 2 spherical vesicles with different areas connect and the integrated closed membrane proceeds shape relaxation. At the onset of the connection, the membrane shape has rotational symmetry for the axis through the centers of the mass of 2 vesicles and the connection point. We assume that the succeeding shape relaxation also holds the rotational symmetry, because some sort of asymmetric force is necessary to violate the rotational symmetry, which is a more complicated problem than can be discussed here.

For further simplification, we only consider membrane shapes that have a constricted *neck* region, and introduce 2 shape parameters: the neck width, $w$, and the area ratio between both sides of the neck, $a$. The constricted neck region is the trace of the connection event and the time evolution of the 2 parameters expresses the approximate deformation processes of 2 vesicle fusions as follows. Just after fusion, $w = 0$ and $a$ is the area ratio between the 2 vesicles. The relaxation of the neck ("explosive" fusion) is expressed by the increase in $w$. The absorption of smaller vesicle by the larger one ("bridge" fusion) is expressed by the decrease in $a$, while $w$ remains small.

Membrane shapes confined by the shape parameters still display a variety. We specify the typical shape of the membrane for a given set of shape parameters $(w, a)$. The typical shape must be more energetically stable than atypical shapes having the same shape parameter values. We choose the shape having the minimum free energy as the typical shape. Then, we obtain the free energy of the typical shape for each set of shape parameters and display the free-energy landscape on the two-dimensional shape parameter space.

### Free energy landscape for small membranes

Suppose a considerably small membrane, which is described by small $A\sqrt{A}$, the free energy is approximated by

$$F \simeq F^{small} = b.$$

To generate rotationally symmetric shapes and to evaluate the bending energy, we consider a curve in the radial and axial space and parametrize it by the function of angle, $\theta$, against the arc-length along the curve, $s$ (*Figure 4—figure supplement 1A*). Rotating the curve around the axis gives a membrane shape. The mean curvature is $\frac{d\theta}{ds} + \frac{sin(\theta)}{r}$ (*Seifert et al., 1991*) and the bending energy for the shape is $b = 2\pi\kappa \int_0^{s_1} \frac{r}{2} \left\{ \frac{d\theta}{ds} + \frac{sin(\theta)}{r} \right\}^2 ds$.

We independently determine the shapes giving the minimum bending energy for the right and left parts of the membrane with an additional boundary condition: a smooth connection between the angles at the boundary ($\theta = \frac{\pi}{2}$ at the boundary).

Here we suppose a membrane with the radius of opening $\bar{w}$, area 1, and give the free energy

$$F = \pi \int_0^{s_1} L\left(\theta, \theta', r, r', \gamma\right) ds$$

With the Lagrange function

$$L = \kappa r \left\{ \theta' + \frac{sin(\theta)}{r} \right\}^2 + \gamma \left( r' - cos(\theta) \right),$$

where $\gamma$ is a Lagrange multiplier. Then the Euler-Lagrange equations for the system are

$$r\frac{d^2\theta}{ds^2} + cos(\theta)\frac{d\theta}{ds} - \frac{sin(\theta)cos(\theta)}{r} - \gamma sin(\theta) = 0; \frac{d\gamma}{ds} = \frac{1}{2}\left(\frac{d\theta}{ds}\right)^2 - \frac{sin^2(\theta)}{2r^2}; \frac{dr}{ds} = cos(\theta).$$

Solving these equations with boundary conditions $(\theta, r)_{s=0} = (0, 0)$, $(\theta, r)_{s=s_1} = \left(\frac{\pi}{2}, \bar{w}\right)$ and $2\pi \int_0^{s_1} r ds = 1$, we get the rotationally symmetric shape having a minimum bending energy $\theta = \theta_{min}(s; \bar{w})$.

We calculated the minimum energy shape for $\bar{w} \in \left[0, 1/\sqrt{2\pi}\right]$, and plotted the bending energy against the radius of the opening. The function is well fitted with a hyperbolic curve

$$\bar{b}(\bar{w}) = 4\pi\kappa \left\{ \frac{\sqrt{\lambda^2\left(\sqrt{2\pi}\bar{w} - 1\right)^2 + 1} - 1}{\sqrt{\lambda^2 + 1} - 1} + 1 \right\}$$

with $\lambda \simeq 7.467$ (*Figure 4—figure supplement 1C*). Thus, we generate the minimum-bending-energy shape for a given set of shape parameters by combining two rescaled partials spheres with scaling ratios $a$ and $1 - a$, and we get the free energy as a function of $(w, a)$

$$F^{small}(w, a) = b(w, a) = \bar{b}\left(\frac{w}{\sqrt{a}}\right) + \bar{b}\left(\frac{w}{\sqrt{1-a}}\right)$$

## Free energy landscape for large membranes

Contrary to the small membrane, a considerably large membrane is described by a large $\Delta P$. We tentatively approximate the free energy of the large membrane with

$$F \simeq F_0^{large} = -\Delta p \cdot A\sqrt{A} \cdot v.$$

The equation indicates that the minimum-free-energy shape corresponds to the shape with the largest volume for the given shape parameters $(w, a)$. Moreover, we independently determine the shapes giving the maximum volume for the right and left parts of the membrane across the constricted neck region, since the area ratio and the size of the boundary (the constricted neck region) are determined by shape parameters. The shape of each part of the membrane corresponds to a partial sphere (*Figure 4—figure supplement 1B*).

A general partial sphere with the radius of opening $\tilde{w}$, area 1, and the volume $\tilde{v}$, is given by

$$\tilde{w} = \alpha sin(\psi)$$

$$1 = 2\pi\alpha^2 \left(1 - \cos\left(\psi\right)\right)$$

$$\tilde{v} = \pi\alpha^3 \left\{ \frac{\cos^3\left(\psi\right)}{3} - \cos\left(\psi\right) + \frac{2}{3} \right\}$$

where, $\alpha$ and $\psi$ are the radius and angle of the partial sphere, respectively (*Figure 4—figure supplement 1B*). Solving the volume as a function of $\tilde{w}$, we get

$$\cos\left(\psi\right) = 2\pi\tilde{w}^2 - 1, \alpha = \frac{1}{2\sqrt{\pi\left(1 - \pi\tilde{w}^2\right)}}$$

$$\tilde{v}\left(\tilde{w}\right) = \frac{1}{8\pi\left(1 - \pi\tilde{w}^2\right)\sqrt{\pi\left(1 - \pi\tilde{w}^2\right)}} \left\{ \frac{\left(2\pi\tilde{w}^2 - 1\right)^3}{3} - 2\pi\tilde{w}^2 + \frac{5}{3} \right\}.$$

We generate the maximum volume shape for a given set of shape parameters by combining two rescaled partials spheres with scaling ratios $\sqrt{a}^3$ and $\sqrt{1-a}^3$, and the volume of the shape is calculated by

$$v\left(w, a\right) = \sqrt{a}^3 \cdot \tilde{v}\left(\frac{w}{\sqrt{a}}\right) + \sqrt{1-a}^3 \cdot \tilde{v}\left(\frac{w}{\sqrt{1-a}}\right)$$

Therefore, we get the free energy as a function of $\left(w, a\right)$

$$F_0^{large}\left(w, a\right) = -\Delta P \cdot \left\{ \sqrt{a}^3 \cdot \tilde{v}\left(\frac{w}{\sqrt{a}}\right) + \sqrt{1-a}^3 \cdot \tilde{v}\left(\frac{w}{\sqrt{1-a}}\right) \right\}$$

The resulting shape of the membrane has a sharply bent region at the neck, which stores large bending energy. Moreover, at the onset of fusion, the osmotic energy has no gradient in the $w$ direction

$$\left. \frac{\partial F_0^{large}\left(w, a\right)}{\partial w} \right|_{w=0} = 0.$$

Thus, the above approximation of neglecting the bending energy for large membranes is not always valid. To draw the time evolution of $w$, we have to incorporate the bending energy as a modification term.

Suppose that the neck region is curved with radius $R_0$ (*Figure 4—figure supplement 1D* and E) which is considerably smaller than the actual neck width $W = \sqrt{A}w$, the bending energy of the neck region is

$$B^{neck} = 2\pi\kappa \int_{\left(\pi-\psi_1\right)r_0}^{\psi_2 r_0} \frac{W - R_0 \sin\left(\xi\right)}{2} \left\{ \frac{\partial\xi}{\partial s} + \frac{\sin\left(\xi\right)}{W - R_0 \sin\left(\xi\right)} \right\}^2 ds,$$

with $s = r_0\theta, \xi = \pi - \theta, \partial\xi/\partial s = \partial\xi/\partial\theta \cdot \partial\theta/\partial s = -1/r_0$, and the volume change by introducing the curved neck region is negligible. The coordinate system of the calculation is shown in *Figure 4—figure supplement 1D* and E. Using the relation $W \gg R_0$, the bending energy becomes

$$B^{neck} \simeq 2\pi\kappa \int_{\pi-\psi_1}^{\psi_2} \frac{W}{2} \left\{ \frac{-1}{R_0} \right\}^2 R_0 d\theta = 2\pi\kappa \frac{W}{R_0} \left( acos\left(\frac{2\pi w^2}{a} - 1\right) + acos\left(\frac{2\pi w^2}{1-a} - 1\right) - \pi \right).$$

Since $R_0$ is assumed to be the minimum curvature radius determined by the physical properties of the membrane, it should be independent from the scale change in size (in other words, the thickness of the neck is fixed under the scale change). Therefore, the bending energy of the neck region depends on the size. The right-hand side of the above equation is decomposed into the size-dependent parameter and the size-independent function of $\left(w, a\right)$,

$$\Gamma = 2\pi\kappa\frac{\sqrt{A}}{R_0}, b^{neck}(w,a) = w\left\{acos\left(\frac{w^2}{a}-1\right) + acos\left(\frac{w^2}{1-a}-1\right) - \pi\right\}.$$

Finally, we get the modified free energy for the large membrane

$$F^{large}(w,a) = \Gamma \cdot b^{neck}(w,a) - \Delta p \cdot A\sqrt{A} \cdot v(w,a).$$

Since $\Gamma \propto \sqrt{A}$, the contribution of the osmotic energy is dominant for the large membrane except the region where the slope of the osmotic energy vanishes.

## Monte-Carlo simulation descending $F^{large}$ landscape

Monte-Carlo simulations on $(w,a)$ space were performed with free energy $F^{large}(w,a)$. We assumed that active fluctuations caused by nonequilibrium processes or activities of motor proteins are modeled by thermal fluctuations with shifted temperature (effective temperature) (*Ben-Isaac et al., 2011*). Although this system has three parameters—the size of the vesicle $\sqrt{A}$, osmotic pressure difference $\Delta p$, and energy of the effective temperature $\Xi$—it is invariant under the transformation $\left(\sqrt{A}, \Delta p, \Xi\right) \rightarrow \left(\lambda\sqrt{A}, \Delta p/\lambda^2, \lambda\Xi\right)$. It means that the essential dimension of the parameters is two. Thus, we chose two rescaled parameters $\Delta p V/\Gamma$ and $\Xi/\Gamma$ with $V = 4A\sqrt{A}/3\sqrt{\pi}$, $\Gamma = 4\pi\kappa\sqrt{A}/R_0$ to be changed. With the initial condition $(w,a) = (0.0, 0.3)$, Monte-Carlo simulations are performed 1000 times for each set of rescaled parameters using the Metropolis method and the frequency of neck expansions in the courses of shape relaxations was calculated. Neck expansion was judged when $w = 0.2$.

