## [Editor Report · eLife assessment]

This study provides **valuable** insights into the role of actin dynamics in regulating the transition of fusion models during homotypic fusion between late endosomes. The evidence supporting the authors' claims is **convincing**. However, while the observations are significant, the study could benefit from further exploration of the mechanistic details and physiological relevance.

---

## [Referee Report · Reviewer #1 (Public review)]

Summary:

This manuscript employs yolk sac visceral endoderm cells as a novel model for studying endosomal fusion, observing two distinct fusion behaviors: quick homotypic fusion between late endosomes, and slower heterotypic fusion between late endosomes and lysosomes. The mathematical modeling suggests that vesicle size critically influences the mode of fusion. Further investigations reveal that actin filaments are dynamically associated with late endosomal membranes, and are oriented in the x-y plane and along the apical-basal axis. Actin and Arf2/3 were shown to appear at the rear end of the endosomes along the moving direction suggesting polymerization of actin may provide force for the movement of endosomes. Additionally, the authors found that actin dynamics regulate homotypic and heterotypic fusion events in a different manner. The authors also provide evidence suggesting that Cofilin-dependent actin dynamics are involved in late endosome fusion.

Strengths:

The unique feature of this study is that the authors use yolk sac visceral endoderm cells to study endosomal fusion. Yolk sac visceral endoderm cells have huge endocytic vesicles, endosomes and lysosomes, offering an excellent system to explore endosomal fusion dynamics and the assembly of cellular factors on membranes. The manuscript provides a valuable and convincing observation of the modes of endosomal fusion and roles of actin dynamics in this process, and the conclusions of the study is justified by the data.

Weaknesses:

While the study offers compelling observations, it falls short in delivering clear mechanistic insights. Key questions remain unaddressed, such as the functional significance of actin filaments that extend apically in positioning late endosomes, the ways in which actin dynamics influence fusion events, and the functional implications of the slower bridge fusion process.

---

## [Referee Report · Reviewer #3 (Public review)]

Summary:

The authors found two endosomal fusion modes by live cell imaging of endosomes in yolk sac lateral endoderm cells of 8.5-day-old embryonic mice and described the fusion modes by mathematical models and simulations. They also showed that actin polymerization is involved in the regulation of one of the fusion modes.

Strengths:

The strength of this study is that the authors' claims are well supported by beautiful live cell images and theoretical models. By using specialized cells, yolk sac visceral endoderm cells, the live images of endosomal fusion, localization of actin-related molecules, and validation data from multiple inhibitor experiments are clear.

Weaknesses:

Although it would be out of scope of this study, there is no experimental verification of whether the mechanism of endosome fusion claimed by the authors occurs in general cells, so the article is limited to showing a phenomenon specific to yolk sac lateral endoderm cells. The methods used were very basic and solid. Most of the image analysis was performed manually, but the results were statistically tested.

Summary:

Seiichi Koike et al. studied two fusion models, explosive fusion, and bridge fusion, utilizing yolk sac visceral endoderm cells. They elucidated these two fusion models in vivo by employing mathematical modeling and incorporating fluctuations derived from actin dynamics as a key regulator for rapid homotypic fusion between late endosomes.

Strengths:

This study uncovered the role of actin dynamics in regulating the transition of fusion models in homotypic fusion between late endosomes and introduced a method for observing the fusion of single vesicles with two different targets.

Weaknesses:

The physiological significance of different fusion models is lacking.

---

## [Author Response]

The following is the authors’ response to the original reviews.

**Reviewer #1 (Recommendations For The Authors):**
While the manuscript provides an interesting observation of the modes of endosomal fusion and roles of actin dynamics in this process and the conclusions of the study are justified by the data, there are concerns regarding the lack of important descriptions or quantification in some of the analyses and additional analyses are needed to strengthen this study. The major issues are outlined below:(1) The authors indicate that Zone 1 is within approximately 1 μm of the apical surface. What are the distances of Zone 2 and Zone 3 from this surface? It would be better if the authors could provide an explanation or hypothesis that explains the early endosomes, late endosomes, and lysosomes are not intermixed but separated along the z-axis.

Thank you for pointing out this important issue. Following the comments, we have added an explanation about the depth of early endosomes, late endosomes, and lysosomes to the text (lines 123-124, 127-128, and 130-131). We have also created a new figure showing their positions in VE cells (Figure 1–figure supplement 1B).

Because endosomes go deeper and mature with repeated fusion and enlargement after endocytosis, early endosomes, late endosomes, and lysosomes are aligned along the z-axis, though the separation is not complete. In confocal microscopic observation, endolysosomal vesicles in VE cells are largely separated into different layers because they are huge and occupy a large space, and as a result, do not exist with much overlap. We have added the explanation to the text (lines 121-122).

(2) The authors compared the size distribution of the late endosomes that underwent fusion with that of the total late endosomes in the observed area 5 min after labeling (Figure 2C). A similar quantification analysis should also be analyzed 15 min after labeling (Figure 3G).

Thank you for the appropriate request. We have added the data showing the size distribution of the late endosomes that underwent fusion at 15 min after labeling, to Figure 3G.

(3) While 3D reconstructions of actin filament patterns under normal conditions are presented (Figures 4 E-F), comparable analyses using cells treated with Cytochalasin D, Jasplakinolide, or S3 peptide needs to be performed.

As requested by the referee, we have performed additional experiments to show 3D reconstructions of actin filaments on late endosomes after pretreatment with cytochalasin D, jasplakinolide, and S3 peptide. We show the data in new figures: Figure 7–figure supplement 1A, Figure 7–figure supplement 2, and Figure 9–figure supplement 1.

(4) The authors should provide a clear description of how they quantified the fusion frequency. Why does the fusion frequency appear very low? Why do Cytochalasin D and jasplakinolide show different effects on heterotypic fusion?

Thank you for pointing out this important issue. We have added the description of how the fusion frequency was quantified to the Materials and Methods (lines 643-645). Briefly, we counted the number of membrane fusion events and the number of late endosomes in the 400-s time-lapse images, and then calculated how many times a single late endosome underwent fusion per minute. The apparent fusion frequency is low because it is expressed in terms of frequency per vesicle per minute.

As for the different effects of cytochalasin D and jasplakinolide on heterotypic fusion, we already discussed this in the manuscript (lines 537-558). In short, actin filaments extending in the apical-to-basal direction are relatively static and late endosomes receive sliding forces along the apical-basal axis by means of myosins (e.g., myosin V and myosin II) in heterotypic fusion. Thus, depolymerization of actin filaments by cytochalasin D treatment reduces heterotypic fusion, and conversely stabilization of actin filaments by jasplakinolide increases heterotypic fusion.

(5) The authors need to analyze the distribution of actin filaments during homotypic fusion post-Cytochalasin D treatment.

As requested by the referee, we have performed additional experiments to show the distribution of actin filaments during homotypic fusion of late endosomes after pretreatment with cytochalasin D. We show the data in a new figure: Figure 7–figure supplement 3.

(6) Clarification is needed on whether overexpressing YFP-Cofilin led to the deterioration of cell functions.

Thank you for the comments. As the reviewer pointed out, overexpression of cofilin can change cellular functions and actin architectures in cells (Aizawa et al., 1997; Popow-Wozniak et al., Histochem. Cell Biol., 2012, (138) 725-36). Although we did not observe apparent morphological changes of VE cells after YFP-cofilin expression, we cannot exclude the possibility that YFP-cofilin overexpression affected the distribution of actin filaments. Therefore, we have described this possibility in the text (lines 425-426).

(7) Although the authors report that the S3 peptide does not affect heterotypic fusion, a reduction in average heterotypic fusion frequency post-treatment was detected (Figure 9G). The authors need to perform a statistical analysis of the quantification performed in Figure 9G.

We apologize for this misleading graph representation. Because S3 peptide treatment did not change the fusion frequency significantly, we simply did not mark statistical significance in the previous graph. To clarify this point, we have added the label “n.s.” (not significant) to Figure 9G.

(8) The authors need to provide the potential functional significance of apically extended actin filaments in positioning late endosomes in the discussion.

We observed 3 different types of actin filaments in the apical region of VE cells (Figure 5). First, the actin mesh in zone 1, which does not interact directly with late endosomes, may function as a barrier preventing enlarged late endosomes from flowing backward from zone 2 to zone 1. Second, actin filaments extending from the apical to the basal direction on the surface of late endosomes are necessary for the movement of late endosomes toward lysosomes in a myosin-dependent manner. Third, the radial branched filaments on the surface of late endosomes temporarily polymerize in an Arp2/3-dependent manner and regulate the lateral movement of late endosomes. This actin organization coordinately regulates the position of late endosomes. We have added this explanation to the Discussion (lines 483-491).

**Reviewer #2 (Recommendations For The Authors):**
(1) What is the effect or physiological significance of the transition in fusion models?

In material transport in cells, explosive fusion that completes membrane fusion quickly is more efficient and physiologically advantageous than slow bridge fusion. On the other hand, larger vesicle size is more effective in membrane trafficking than smaller size because large vesicles can transport a large amount of cargo molecules. However, as our mathematical modeling predicts, an increase in vesicle size leads to bridge fusion and decreases the transportation rate. Actin forces can resolve these conflicting effects because they convert the fusion mode from bridge to explosive in late endosomes in VE cells

(2) I am confused about how to study heterotypic fusion between late endosomes and lysosomes using only transferrin labeling.

We are sorry for any confusion this may have caused. Indeed, at first, we discovered that late endosomes shrank and disappeared after labeling of endocytic vesicles with transferrin only (Figure 3A). However, subsequently, we speculated that this disappearance was the result of heterotypic fusion with lysosomes, and to prove this possibility, we developed a double-labeling method in which late endosomes and lysosomes were labeled with 2 different colors (Figure 3B). In short, VE cells were incubated with dextran rhodamine for 20 min and subsequently pulse-labeled with Alexa Fluor 488-labeled transferrin for 5 min: when VE cells were observed, dextran rhodamine was already transported to lysosomes, whereas Alexa Fluor 488-labeled transferrin was still present in late endosomes, enabling the two vesicles to be observed separately.

**Reviewer #3 (Recommendations For The Authors):**
(1) It is concerning that there are several points that are not fully explained regarding microscopic image analysis.(a) How were zones 1, 2, and 3 defined and how were the zones determined at each observation? Did the authors determine the zones subjectively based on the approximate size of the vesicles and the passage of time, or statistically by measuring endosomes from images of whole cells? The authors should describe this and also provide the approximate z-directional thickness of each of zones 1, 2, and 3.

Thank you for pointing out this important issue, which is also raised by Reviewer #1. We initially analyzed the distribution and size of early endosomes, late endosomes, and lysosomes in VE cells by use of vesicle-specific markers (Figure 1B). Thereafter, at each observation, we determined the zones based on the characteristic size of the vesicles and time after labeling of endocytic vesicles. Especially, images of zone 2 and zone 3 were taken by focusing on the z-axis where late endosomes and lysosomes occupied the largest area in the optical slice images, respectively (lines 636-639). As for the z-directional thickness of each zone, we have added a description to the text (lines 123-124, 127-128, and 130-131) and also created a new figure showing their positions in VE cells (Figure 1–figure supplement 1A).

(b) Regarding "vesicle size" measured from confocal microscopy images: Does "vesicle size" mean surface area or maximum cross-sectional area? In any case, the authors should describe how and what area of the vesicles was measured from the images. The mathematical model used the surface area of the vesicle as a parameter. Better to be consistent.

Thank you for the important questions. As the reviewer pointed out, the cross-sectional area of endosomes varies depending on the focal plane. To ensure uniformity of the focal plane across different images, we took the images by focusing on the z-axis where late endosomes (zone 2) or lysosomes (zone 3) occupied the largest area in the image. In the focal plane, we measured the size of all intact, unfragmented endosomes. We have now added this explanation to the Method section (lines 636-639).

(c) The authors showed several time-lapse imaging data without a description of what "0 s" is the starting time of. For example, "0 s" in Figures 2A, B, 3A, and B, may have different meanings. Other data should be carefully examined and described.

We apologize for the inadequate description. As the reviewer pointed out, each panel has a different meaning of "0s."Therefore, we have added explanation of the meaning of “0s” to the relevant figure legends (Figure 2A, B; Figure 3A, B; Figure 6A, F; Figure 7A, E, F; Figure 8A, Figure 6–figure supplement 1C, Figure 7–figure supplement 1B, Figure 7–figure supplement 3, Figure 7–figure supplement 4).

(d) The meaning of "fusion time" in Figures 2D and 3F is unclear. Although it was speculated that the authors estimated it from the change in shape of the vesicles, how it was measured should be described.

We apologize for the inadequate description. To indicate more clearly, we have added an explanation of the "fusion time" to the legend of Figures 2D and 3F (lines 898-899 and line 923, respectively).

(2) The structure of the paragraph starting on line 158 is inappropriate. The authors state in line 159 that "this disappearance appeared to result from fusion of late endosomes with the underlying lysosomes". However, no hetero-fusion was observed here, only the disappearance of vesicles. The authors should mention that hetero-fusion occurred only after analysis of Figure 3CD.This reviewer thinks it is natural to state in this order: first, the disappearance of transferrin-positive vesicles was observed (Figure 3A). Such vesicles became dextran-positive as the transferrin signal began to disappear (Figures 3 B ,C, D). Thus, this is thought to indicate that hetero-fusion has occurred.

We agree with the reviewer's comment and have rewritten the text following the reviewer's suggestion (lines 163-165, 176-180).

(3) The mathematical model estimated that the vesicle size of 0.22-1.0 [𝜇𝑚2] is the size to switch the fusion mode. Since this is close to the size of endosomes in general cells, the authors may be able to discuss the generality of the fusion mode theory. It is up to the author to respond to this suggestion or not.

Thank you for the comments. As our mathematical model depends on the assumption that the osmotic pressure is constant, late endosomes in VE cells, exhibiting a swollen morphology, may have higher osmotic pressure compared with endosomes in other cells and if so, the predicted vesicle size when the fusion mode switches may differ. Thus, we have decided not to mention the relationship between the vesicle size and fusion mode switching.

(4) In Line 302 the authors mentioned "These results indicated that actin spots on the surface of late endosomes were dynamically regulated, especially in the apical area." However, the t-halves of 11.5s and 18.9s are only slightly different and of the same order, so it would be too much to say that dynamic regulation of actin occurs specifically in the apical region from a difference of this magnitude. The authors should weaken their arguments. It would be good to do a statistical test for significance between the FRAP data.

Thank you for pointing out this important issue. To highlight the significant difference in the FRAP assay, we have added a new panel showing the statistical analysis of the halftime of recovery of each region of VE cells (Figure 6E). These data indicate that a significance difference in the halftime of recovery (t1/2) between actin spots in the apical and basal regions of zone 2. However, following the reviewer’s comment, we have weakened the description of the FRAP assay results (lines 310-312).

(5) The discussion section is rather redundant. It could be shortened to be more concise instead of repeating the results.

Thank you for the comments. We have shortened the Discussion section.

Minor commentsIn Figure 2C, the statistical test method was not described in the legend.

Thank you for the comments. We have added the data of the statistical test to the figure legend of Figure 2C (lines 895-896).

Figure 3G does not look like a normal distribution, so the t-test is inappropriate.

Thank you for the comments. We have changed the statistical analysis method and used the Mann-Whitney U test. For the same reason, we have changed the analysis method shown in Figure 2C.

Is Figure 5D the image of zone 1 because it is close to the apical plane? If so, are the IgG-positive structures early endosomes rather than late endosomes? This seems inconsistent with the data in Figure 1.

Thank you for the comments. The round vesicles observed in this panel are the late endosomes in zone 2. Because most of the internalized fluorescence marker has moved to the late endosomes in zone 2 at this time point (5 min after chasing), early endosomes are not labeled in this image. We have added a dotted line to the x-z axis image (the second top panel) to indicate the depth of the x-y axis image (top panel) in Figure 5D.

Figure 6B appears to have little or no fluorescence recovery. Is this a typical example? It is also unclear if this is an apical or basal example.

Thank you for the comments. This image is a typical example. We focused on the dot structures on the surface of late endosomes rather than the fluorescence intensity over the entire photobleached area. To prevent misunderstanding, we have added arrowheads to highlight the actin dot structures that we were analyzing. The FRAP data shown in Figure 6B were obtained at the apical region of zone 2. We have also added this information to the figure legend.